# New ground-based Fourirer-transform near-infrared solar absorption measurements of $XCO_2$, $XCH_4$ and XCO at Xianghe, China

Yang Yang[1,3,2], Minqiang Zhou[2], Bavo Langerock[2], Mahesh Kumar Sha[2], Christian Hermans[2], Ting Wang[1,3], Denghui Ji[1,3], Corinne Vigouroux[2], Nicolas Kumps[2], Gengchen Wang[1,3], Martine De Mazière[2], and Pucai Wang[1,3]

[1]LAGEO, the Institute of Atmospheric Physics, Chinese Academy of Sciences, Beijing, China
[2]Royal Belgian Institute for Space Aeronomy, Brussels, Belgium
[3]University of Chinese Academy of Sciences, Beijing, China

**Correspondence:** Minqiang Zhou (minqiang.zhou@aeronomie.be), Pucai Wang (pcwang@mail.iap.ac.cn)

**Abstract.** The column-averaged dry-air mole fractions of $CO_2$ ($XCO_2$), $CH_4$ ($XCH_4$) and CO (XCO) have been measured with a Bruker IFS 125HR Fourier transform infrared spectrometer (FTIR) at Xianghe (39.75 °N, 116.96 °E, North China) since June 2018. This paper presents the site, the characteristics of the FTIR system and the measurements. The instrumental setup follows the guidelines of the Total Carbon Column Observing Network (TCCON): the near-infrared spectra are recorded by an InGaAs detector together with a $CaF_2$ beam splitter, and the HCl cell measurements that are recorded regularly to derive the instrument line shape (ILS) showing that the instrument is correctly aligned. The TCCON standard retrieval code (GGG2014) is applied to retrieve $XCO_2$, $XCH_4$ and XCO. The resulting time series between June 2018 and July 2019 are presented, and the observed seasonal cycles and day-to-day variations of $XCO_2$, $XCH_4$ and XCO at Xianghe are discussed. In addition, the paper shows comparisons between the data products retrieved from the FTIR measurements at Xianghe and co-located Orbiting Carbon Observatory-2 (OCO-2) and Tropospheric Monitoring Instrument (TROPOMI) satellite observations. The comparison results appear consistent with validation results obtained at TCCON sites for $XCO_2$ and $XCH_4$, while for XCO they highlight the occurrence of frequent high-pollution events. As Xianghe lies in a polluted area in North China where there are currently no TCCON sites, this site can fill the TCCON gap in this region and expand the global coverage of the TCCON measurements. The Xianghe FTIR $XCO_2$, $XCH_4$ and XCO data can be obtained at https://doi.org/10.18758/71021049 (Yang et al., 2019).

## 1 Introduction

The rapid economic growth of China has contributed to 28.5% of the global total carbon dioxide ($CO_2$) emissions from fossil fuel consumption and cement production (Jackson et al., 2017). China dominates global $CO_2$ fossil emissions with an average increase of $3.8\% yr^{-1}$ between 2008 and 2017 (Le Quéré et al., 2018). In addition, 14% to 22% of the global anthropogenic methane ($CH_4$) emissions in the 2000s were attributed to China (Kirschke et al., 2013). It is clear that China should play an important role in the reduction of global carbon emission and climate change mitigation. A decreasing linear trend of -0.41 $\pm$ 0.09%$yr^{-1}$ in carbon monoxide (CO) concentrations from 2005 to 2016 has been observed over East Asia and 76% of

this decrease is due to the CO emission control in China (Zheng et al., 2018). However, the estimation of Chinese carbon emissions still have large uncertainties, ranging from $\pm$ 5% to $\pm$ 10% (Gregg et al., 2008; Le Quéré et al., 2018; Andres et al., 2012). A good understanding of carbon emissions requires accurate monitoring of $CO_2$, $CH_4$, CO and other direct and indirect greenhouse gases.

$CO_2$ is the most important anthropogenic greenhouse gas with a radiative forcing of $1.82 \pm 0.19$ W/m$^2$ in 2013 (IPCC, 2013). The globally averaged surface dry-air mole fraction of $CO_2$ increases steadily in the atmosphere from 278 ppm pre-industrial level and has reached up to $405.5 \pm 0.1$ ppm in 2017 (WMO, 2018). The annual increasing rate of $CO_2$ in the atmosphere during the last 10 years is 2.24 ppm/year (WMO, 2018). The enhancement of $CO_2$ is primarily caused by human activities, such as the fossil burning and the land-use change (Peters et al., 2012). Atmospheric $CH_4$ is the second important

anthropogenic greenhouse gas, and its globally averaged surface dry-air mole fraction increased from pre-industrial of 722 ppb up to $1859 \pm 2$ ppb in 2017 (WMO, 2018), with an average growth rate of 7 ppb/year during the last decade. Although the $CH_4$ abundance is much lower than that of $CO_2$, the comparative impact of $CH_4$ is about 28 times greater than $CO_2$ over a 100-year period (IPCC, 2013). It is reported that the radiative forcing of $CH_4$ has increased to $0.48 \pm 0.05$ W/m$^2$ in 2013 (IPCC, 2013). 50-65% of $CH_4$ is released from human activities such as energy production/consumption, industry, agriculture,

biomass burning, and waste management activities and another 40% from natural emissions (IPCC, 2013). Atmospheric CO is an indirect greenhouse gas and is mainly emitted from fossil fuel combustion and biomass burning (Yin et al., 2015).

     The Total Carbon Column Observing Network (TCCON) uses ground-based FTIR spectrometers to measure the direct solar radiation in the near infrared spectral region, from which the total column-averaged dry-air mole fractions of $CO_2$, $CH_4$, $N_2O$, CO, HF, $H_2O$ and HDO are retrieved (Wunch et al., 2011b). Because of their relatively high precision and accuracy,

TCCON data are widely used in satellite validations and model comparisons (Zhou et al., 2016; Ostler et al., 2016; Crisp et al., 2017; Borsdorff et al., 2018; Velazco et al., 2019). Today, there are 25 active TCCON sites (https://tccon-wiki.caltech.edu/) covering the latitude band from 80 °N to 45 °S. Most TCCON sites are in North American, Europe, East Asia (South Korea and Japan) and Oceania. The Hefei station, located in Eastern China, is the first Chinese site that will potentially join the TCCON network. In 2016, a FTIR Bruker IFS 125HR instrument was installed at Xianghe (39.75 °N, 116.96 °E, 30m a.s.l.) and started

observations following the TCCON settings in June 2018. As there are no TCCON sites in North China, the FTIR at Xianghe aims to fill the gap in the network in this region.

     The Japanese Greenhouse gases Observing Satellite (GOSAT) was successfully launched in 2009 by the Japan Aerospace Exploration Agency (JAXA). It is the first spacecraft to measure atmospheric $CO_2$ and $CH_4$ with high-resolution spectra at SWIR wavelength (Kuze et al., 2009). The Orbiting Carbon Observatory-2 (OCO-2) was launched on 2 July 2014 by NASA

and is devoted to enhancing our understanding of regional scale $CO_2$ exchanges between the surface and the atmosphere (Crisp et al., 2004; Eldering et al., 2017; Crisp et al., 2017). The Tropospheric Monitoring Instrument (TROPOMI) was launched by ESA on 13 October 2017 as the single payload of the Sentinel-5 Precursor (S5P) satellite. It aims at providing accurate and timely observations of abundances of atmospheric species, such as $CH_4$ and CO, for air quality and climate change research and services (Borsdorff et al., 2018). However, previous validation work (Wunch et al., 2017; Lambert et al., 2019) based on

FTIR measurements has no study in North China due to the absence of TCCON sites in this area, so that it is important to add Xianghe site for evaluation of satellite products in this area.

In this paper, we describe the ground-based FTIR system at Xianghe, with a focus on the measurements of atmospheric $CO_2$, $CH_4$ and CO (Yang et al., 2019). The column-averaged dry-air mole fractions of these gases are retrieved by the GGG2014 (Wunch et al., 2015) code in the period between 14 June 2018 and 19 July 2019. The paper is structured as follows. Section 2 introduces the Xianghe site and the FTIR system. In Section 3, the retrieval and filtering methods are described. The time series of $XCO_2$, $XCH_4$ and XCO are shown and discussed. In the next Section, the OCO-2 ($XCO_2$) and TROPOMI ($XCH_4$ and XCO) satellite observations are compared with the FTIR measurements at Xianghe. Finally, conclusions are drawn in Section 5.

## 2 FTIR measurements

### 2.1 Location and experimental set-up

Xianghe site has been operated by the Institute of Atmospheric Physics, Chinese Academy of Sciences since 1974. The site is about 50 km to the east-southeast of Beijing and 70 km to the north-northwest of Tianjin (see Figure 1, right panel). The Xianghe county acts as an integrated transportation and transfer center in Beijing-Tianjin-Hebei region, which is one of the most populous and economically dynamic areas in China (Ran et al., 2016). Xianghe has a middle latitude monsoon climate, with a prevailing south-east wind in summer and a north-west wind in winter (Song et al., 2011). The maximum temperature at Xianghe site is around 38 °C in summer, and the minimum temperature is around -10 °C in winter. The raining days occur mainly in summer including some days with extreme precipitations larger than 100 mm/day.

The Bruker IFS 125HR instrument was installed in the upper level of a four-story building in June 2016. About 2 years later, in June 2018, a solar tracker was installed on the roof (50 m a.s.l.), to guide the direct solar radiation into the FTIR instrument. The distance between the solar tracker and the entrance window of the FTIR instrument is about 3 m. The solar tracker uses a camera inside the IFS 125HR spectrometer to ensure that the center of the solar disk always focuses on the entrance aperture of the spectrometer, with an active feedback loop. This system is set up following the developments from Neefs et al. (2007) and Gisi et al. (2011). The FTIR operates only under clear-sky daytime conditions. A rain sensor and a solar irradiation (both total and direct) sensor, are installed next to the solar tracker, to monitor the weather conditions and to control the opening and closing of the solar tracker hatch. To protect the mirrors (Aluminum, coating with $MgF_2$) of the solar tracker, the hatch of the tracker automatically closes under rainy conditions and during nighttime. A heating system is operated in the tracker system to keep the temperature of the rotary stages and the mirrors close to 15°C in winter. Inside the lab, the air-conditioning keeps the room temperature stabilized around 25 °C.

The near infrared (NIR) spectra are recorded by a indium gallium arsenide (InGaAs) detector and the middle infrared (MIR) spectra are recorded by a liquid nitrogen cooled indium antimonide (InSb) detector. The entrance window and the beamsplitter are made of $CaF_2$. The spectral ranges of the NIR and MIR spectra are 3800-11000 $cm^{-1}$ and 2000-5000 $cm^{-1}$, respectively. The InSb detector at Xianghe records spectra in the AC mode. The InGaAs detector at Xianghe was operated in the AC

mode before 31 May 2019, but since then in the AC + DC mode to be compliant with TCCON standards. The spectrometer settings automatically alternate between NIR and MIR measurements during each clear-sky day. The entrance aperture of the spectrometer in the NIR spectrum is set to 0.5 mm and changed to 0.8 mm after 19 June 2019. There are approximately 70 NIR InGaAs spectra for each clear day. The InGaAs spectra are recorded with a maximum optical path difference (MOPD) of 45 cm, corresponding to a spectral resolution of 0.02 cm$^{-1}$. Each measurement contains 2 scans (one forward and one backward), taking about 145 s.

About 50 m south-west of the laboratory building, a weather station is operated on a 110 m-height tower, at 62 m above the ground, measuring pressure, temperature, humidity, wind direction and wind speed. The pressure sensor is located inside the LI-7550 Analyzer, which has an accuracy of 1 hPa. On 30 May 2019, a new weather station was installed at the same height as the solar tracker. The distance between the weather station and the solar tracker is about 2 m. The pressure sensor is PTB210A Digital Barometer, with an accuracy of 0.07 hPa.

## 2.2 Instrument line shape

The instrument line shape (ILS) reflects the performance and alignment of the instrument (Schneider et al., 2008), which might be distorted by the shear or angular misalignment of the instrument or the field of view (FOV) (Hase et al., 1999; Wunch et al., 2015). A perfectly-aligned interferometer will perfectly center the Haidinger fringes on the field stop at all optical path differences (OPD). The offset between the moving cube-corner retro-reflector (CCRR) and the fixed CCRR will cause the Haidinger fringes moving away from the center when the mirror moves away from zero optical path difference (ZOPD) and this is called shear misalignment. The angular misalignment is caused when the IR beam is not parallel to the rails. At Xianghe 2 HCl cell spectra are recorded every day after sunset. The modulation efficiency amplitude (ME) and phase error (PE) are retrieved by LINEFIT14.5 using 13 HCl microwindows under non-vacuum status (Hase et al., 1999). The degree of freedoms for signal (DOFS) of apodization and phase are about 4.1 and 4.2, respectively. The ME is derived from the ratio of the misaligned fringe amplitude to the theoretical fringe amplitude. LINEFIT14.5 normalizes ME to be 1.0 at ZOPD. Figure 2 shows the ME and PE along with the OPD, together with the maximum ME loss and maximum PE deviation at Xianghe. The ME has a maximum loss at the MOPD while PE has a maximum deviation at about 20 cm (positive value) or the MOPD (negative value). Note that the Tungsten lamp used to measure the HCl cell spectra broke down in February 2019. According to the TCCON requirement, ME changes must be within 5% at MOPD and PE less than 0.02 rad (Wunch et al., 2011b). The mean of the maximum ME loss is $0.022 \pm 0.004$, and the mean of the maximum absolute PE deviation is $0.014 \pm 0.003$ rad. Only few days in September 2018 have maximum absolute PE slighly larger than 0.02 rad. In general, the alignment of the instrument slightly declines over time, but the ME and PE remain compliant with the TCCON requirements during the whole time period. A sensitivity study performed by Hase et al. (2013) showed that the uncertainty in XCO$_2$ is about 0.035% (0.14 ppm) for a ME change of 4%, which is within the 0.8 ppm (SZA less than 80 °) estimated retrieval accuracy of TCCON XCO$_2$. Since the ME of the FTIR instrument at Xianghe is about 2-3%, the uncertainty from the ILS on the greenhouse gas retrievals can be ignored.

## 2.3 Signal-to-noise ratio

The time series of the signal-to-noise ratio (SNR) of the InGaAs spectra at Xianghe is shown in Figure 3. The SNR (Eq.1) is calculated as the ratio between the maximum intensity ($max(I)$) of the spectrum in the spectral range of 3800-11000 cm$^{-1}$ and the noise level. The standard deviation of the intensity between 2350 and 2450 cm$^{-1}$ ($STD(noise)$) is calculated as the noise level, since no signal is recorded in this window.

$$SNR = max(I)/STD(noise). \tag{1}$$

There are no measurements between 7 July and 22 August 2018 due to a power cut. The SNR decreases quickly with time because Xianghe is located in a polluted area (Robert and Richard, 2015; Li et al., 2007) causing rapid degradation of the mirrors of the solar tracker (Feist et al., 2016). In order to obtain a high SNR, the mirror of the solar tracker was cleaned on 14 November 2018 (first yellow line in Figure 3), with the SNR increased from 300 to 500. However, the SNR decreased back to the level of 300 about 3 weeks later, probably because of the increased level of air pollution and relatively lower solar irradiation in winter. The polluted mirror of the solar tracker was replaced with a new one on February 1, 2019 (second yellow line in Figure 3), enhancing the SNR from about 300 to 450. With the DC signal recording (first red line in Figure 3), between May 31 2019 and June 19 2019 the SNR dropped quickly below 200. Therefore, the aperture size was increased from 0.5 to 0.8 mm on 19 June 2019 (second red line in Figure 3) and the mirror was cleaned again on 25 June 2019 (third yellow line in Figure 3), making the SNR rise again above the level of 300.

## 3 FTIR retrievals

### 3.1 Retrieval Methodology

The non-linear least-squares fitting code GGG2014 (Wunch et al., 2015) is used to retrieve XCO$_2$, XCH$_4$, XCO and some other gases from the NIR solar absorption spectra at Xianghe.

$$TC_r = TC_a + \boldsymbol{A}(\boldsymbol{x_t} - \boldsymbol{x_a}) + \varepsilon, \tag{2}$$

where $TC_a$ and $TC_r$ are the a priori and retrieved total columns, $\boldsymbol{A}$ is the column averaging kernel, $\boldsymbol{x_t}$ and $\boldsymbol{x_a}$ are true and a priori partial column profiles, $\varepsilon$ is the uncertainty. In the forward model, there are 70 equidistant 1km thick layers from the mean sea level up to 70 km altitude. The a priori profiles of gases are generated by an empirical model based on surface in situ, Atmospheric Chemistry Experiment - Fourier Transform Spectrometer (ACE-FTS) and the Jet Propulsion Laboratory MkIV (Mark IV) interferometer measurements. The inter-annual trends and seasonal variations of the species are taken into account, and the a priori profiles are adjusted based on the local tropopause pressure at local noon (Toon and Wunch, 2015). The temperature, pressure and water vapour profiles are taken from the National Centres for Environmental Prediction (NCEP) reanalysis data (Kalnay et al., 1996). The surface pressure, temperature and water vapour are from the local weather station.

The column averaging kernel represents the vertical sensitivity of the retrieved total column to the true partial column profile. The typical averaging kernels of XCO$_2$, XCH$_4$ and XCO are shown in Figure 4 of Wunch et al. (2011a). In general, the

retrieved $CO_2$ and $CH_4$ total columns have good sensitivity in the troposphere and the stratosphere. The retrieved CO column underestimates a deviation from the a priori partial column in the troposphere but overestimates a deviation from the a priori partial column in the stratosphere.

The retrieval windows of $CO_2$, $CH_4$, CO and $O_2$ are listed in Table 1 in Wunch et al. (2010). As an example, Figure 4 shows the residuals of the spectral fitting for $CO_2$ and $O_2$ for one NIR spectrum at a SZA of 22.9 °at Xianghe. The root mean square of the residuals for $CO_2$ in the spectral window 6180-6260 cm $^{-1}$, $CO_2$ in the window 6297-6382 cm$^{-1}$ and $O_2$ in the window 7765-8005 cm$^{-1}$ are 0.24%, 0.25% and 0.37%, respectively, which compare well to the results in Toon et al. (2009).

The spectroscopy is the ATM line list (Toon, 2017). The total column-averaged dry-air mole fraction of gas ($X_{gas}$) is then derived from the ratio between the retrieved total column of the target species and the retrieved total column of $O_2$ and from the dry-air mole fraction of $O_2$ (0.2095)

$$X_{gas} = 0.2095 \times column_{gas}/column_{O_2}. \tag{3}$$

Using the ratio between the target species and $O_2$ reduces the uncertainties common to both gases, e.g., the surface pressure, water vapour, solar tracker pointing and zero level offsets. In addition, a post-processing of the results is done: (1) the airmass dependence of the retrieval results which is known to be an artifact caused by spectroscopic uncertainties, is reduced by applying an empirical airmass-dependent correction, and (2) a constant scaling factor for each gas is applied to calibrate the TCCON measurements to the WMO scale (Wunch et al., 2015, 2010).

## 3.2  Data quality control

As the recording time for one InGaAs spectrum takes about 145 s, the stability of the incoming solar intensity during this period is important for the quality of the spectrum (Beer, 1992). If there are clouds or heavy aerosols in the light path between the FTS and the sun during the spectrum recording, the fractional line depth in FTIR spectra will be distorted (Ridder et al., 2011; Keppel-Aleks et al., 2011). To select the good quality spectra, we have discarded all the spectra with SNR less than 200.

The DC correction can be used to remove the solar irradiation variation in each spectrum (Keppel-Aleks et al., 2007). To filter out spectra affected by the occurrence of clouds or high aerosol load before May 31, 2019 (in AC recording mode), we use the variations of the direct solar irradiation observed from the solar irradiation detector installed close to the solar tracker system (see Section 2.1). The solar direct irradiation is recorded every 3 seconds providing us with about 25 measurement points per forward or backward scan. If there are no clouds or strong loads of aerosol during the scan, the direct solar irradiation should remain relatively stable. We have classified measurement points during the scan as problematic if the solar intensity is lower than a certain percentage threshold $\beta$ of the maximum intensity. A spectrum is selected as a good one if the number of problematic measurement points is smaller than or equal a certain percentage threshold $\gamma$ of the total number of measurement points during the measurement. We tested the filtering method with $\gamma$ values of 0%, 5% and 10%, and $\beta$ values of 85%, 90% and 95% on all InGaAs spectra (about 12000) from 14 June 2018 to 31 December 2018. To estimate the precision of the retrievals at Xianghe, we limit the windows over which standard deviations (STD) are calculated to a smaller range of solar zenith anglee (SZA less than 30°) to reduce the potential influences, such as a diurnal cycle, the influence of local sources in

such a heavily urbanized area and a residual airmass dependence of the retrievals as pressure broadening is not accounted for in the GGG2014 spectroscopy. We select all the days when at least 5 measurements are available. The results are shown in Table 1. The STDs of $XCO_2$, $XCH_4$ and XCO decrease with increasing $\beta$ or decreasing $\gamma$ thresholds. The $\beta$ and $\gamma$ values are selected mainly based on the $XCO_2$ data. According to Pollard et al. (2017), the precision target for TCCON is 0.1% ($\sim$0.4

ppm) for $XCO_2$ to meet the model requirement (Olsen and Randerson, 2004). However, the precision of the TCCON $XCO_2$ is estimated to be 0.2% ($\sim$0.8 ppm) based on the perturbation of the GGG2014 inputs (Wunch et al., 2015). In order to reduce the variation of the measurements and to keep as many as useful measurements, we choose the $\beta$=90%, $\gamma$=0% as the criteria for the solar intensity filtering. The mean STD of $XCO_2$, $XCH_4$ and XCO are 0.496 ppm, 0.0035 ppm and 2.559 ppb, respectively, and there are 84.7% spectra remained. Note that the mean STD of $XCO_2$ is 0.12%, which is slightly worse than the target of

the TCCON $XCO_2$ precision of 0.1%, but it is better than the estimated uncertainty of 0.2%. To evaluate the precision of the retrievals at Xianghe, we compare the STD of $XCO_2$ measurements at Xianghe with several TCCON sites with similar latitude (Lamont, Karlsruhe, Pasadena, Rikubetsu, Tsukuba, Saga, Orleans and Anmeyondo), and the STD of $XCO_2$ at Xianghe is comparable with these sites.

The 110 m-height meteorology tower is collinear with the solar irradiation sensor and the sun tracker. It generates a shadow

on the mirror of the tracker system, affecting the spectra in the afternoon. The duration of the shadow is about 8-15 minutes, corresponding to about 2-3 InGaAs measurements. The shadow occurs at around 6:30 UTC (14:30 LTC) in summer and around 9:30 UTC (16:30 LTC) in winter. The upper panel in Figure 5 shows the solar direct intensity from 1 October 2018 to 3 October 2018, with a zoom in over the time period between 6:36 and 7:02 (UTC) on 2 October 2018 when the tower shadow is observed by the FTIR (see the inside camera image in the middle panel). The affected $XCO_2$ retrieval during the shadow appearing time

are displayed in the bottom panel. The spectra sampling time is recorded at time of ZOPD. The blue dots in the upper panel denote the SNR of spectra filtered out with solar intensity (SI), the cyan dots denote the SNR of spectra filtered out with SNR, and the red dots denote the SNR of spectra after both filtering. All of these SNR values are also zoomed in the middle panel of Figure 5. It is clear that the spectra polluted by the tower shadow can be filtered out by the combination of SNR filtering and SI filtering with $\beta = 90\%$ and $\gamma = 0\%$ (see cyan and blue dots in middle panel, appearing in the same period with the shadow).

The other blue dots in the upper panel appearing outside the shadow shows that SI filtering can also filter out spectra which are polluted by clouds.

Figure 6 shows the time series of $XCO_2$ with and without filtering between 14 June 2018 and 19 July 2019. In the bottom panel, the cyan dots denote all the $XCO_2$ values without any filtering, the blue dots denote the $XCO_2$ values after SNR filtering applied and the red dots denote the $XCO_2$ values after both SNR and SI filtering applied. In the upper panel, the blue dots

denote the daily STDs of $XCO_2$ when only the SNR filtering is applied, while the red ones denote the daily STDs of $XCO_2$ when both the SNR and the SI filtering are applied. Based on the measurements with the SZA less than 30°, Table 2 shows that SNR filtering apparently reduces the average daily STDs both before (from 1.19 ppm to 0.94 ppm) and after (from 1.06 ppm to 0.53 ppm) DC signal recording. It is clear that before DC recording, many $XCO_2$ outliers are removed by the SI filtering, with the average daily STD of $XCO_2$ decreasing from 0.94 ppm to 0.57 ppm. However, the SI filtering makes the STD of

$XCO_2$ increases from 0.53 ppm to 0.88 ppm for the AC+DC period. The main reason is that many retrievals corrected by the

DC correction procedure are still filtered out with SI. Therefore, for the period with AC+DC mode, only the SNR filtering is applied.

Poor instrument alignment, spectral ghost, error in the time assigned to the spectrum or faulty pressure sensor may cause a dramatic jump in $X_{air}$ (Washenfelder et al., 2006; Wunch et al., 2011a). The retrieved $X_{air}$ (after filtering) are shown in Figure 7 to confirm the good quality of the retrievals. $X_{air}$ is defined as

$$X_{air} = \frac{TC_{dry,air}}{TC_{O_2}/0.2095} = \frac{0.2095}{TC_{O_2}}\left[\frac{P_s}{gm_{dry,air}} - TC_{H_2O}\frac{m_{H_2O}}{m_{dry,air}}\right]. \tag{4}$$

where $m_{dry,air}$ and $m_{H_2O}$ are the molecular mass of dry air and water vapour and g is the column-averaged gravitational acceleration, $P_s$ is the surface pressure and $TC_{dry,air}$, $TC_{O_2}$ and $TC_{H_2O}$ are total columns of dry air, $O_2$ and $H_2O$, respectively. The $X_{air}$ is around 0.98 due to a ~2.0% bias in the $O_2$ spectroscopy (Kivi and Heikkinen, 2016). Figure 7 shows that the $X_{air}$ at Xianghe all pass TCCON standard quality check (between 0.96 and 1.04) and is stable over time with a mean value of 0.982 and a STD of 0.003.

### 3.3 Retrieval results and discussions

The time series of $XCO_2$, $XCH_4$ and XCO after both SNR and SI filtering from June 2018 to July 2019 are shown in Figure 8. The monthly mean of $XCO_2$, $XCH_4$ and XCO at Xianghe, Pasadena (34.1°N ) (Wennberg et al., 2014), Lamont (36.6°N) (Wennberg et al., 2016) and Karlsruhe (49.1°N) (Hase et al., 2014) from June 2018 are also displayed in Figure 9. Based on these measurements, the seasonal variations and day-to-day variations of $XCO_2$, $XCH_4$ and XCO are assessed. $XCO_2$ is low in summer and high in winter and spring at Xianghe, with maximum monthly mean $414.27 \pm 0.94$ ppm in April and minimum monthly mean $401.58 \pm 1.32$ ppm in August. This seasonal behaviour is similar to those at Pasadena, Lamont and Karlsruhe, which are located in the northern mid-latitude zone. The peak-to-peak amplitude of the seasonal variation of $XCO_2$ is 12.41 ppm, which is larger than 7.45 ppm at Pasadena, 7.78 ppm at Lamont and 7.98 ppm at Karlsruhe. $XCH_4$ is low in spring and high in autumn and summer, with a maximum monthly mean of $1.893 \pm 0.017$ ppm in August and a minimum monthly mean of $1.857 \pm 0.014$ ppm in March. It is found that the seasonal cycle of the $XCH_4$ at Xianghe is very different with other sites at a similar latitude, as the observations at the other 3 stations show low values in summer and high values in autumn and winter. The peak-to-peak amplitude of the $XCH_4$ seasonal variation is about 0.039 ppm at Xianghe, which is also larger than 0.029 ppm at Pasadena, 0.028 ppm at Lamont and 0.024 ppm at Karlsruhe. XCO at Xianghe is relatively high during the whole year. The background value of XCO is about 90 ppb, and the high XCO measurements can reach up to 200 ppb. The monthly mean XCO values at Xianghe are always higher than those at the other 3 stations, indicating that regional pollution sources are frequently observed at Xianghe.

Similar day-to-day variations are observed among $XCO_2$, $XCH_4$ and XCO. The high values are related to the local emissions while the low values are influenced by the air transported from remote places. In this section, we use CO as a trace gas to evaluate the correlations between XCO and $XCO_2$, XCO and $XCH_4$. Figure 10 (a, b) shows the correlations between the XCO and $XCO_2$ daily means and between the XCO and $XCH_4$ daily means at Xianghe. $XCO_2$ is high in winter and low in summer, and $XCH_4$ is high in summer and autumn and low in winter. In order to reduce the impact from their seasonal variations, a

linear regression model is used to fit the time series of measurements

$$Y(t) = A_0 + \sum_{k=1}^{3}(A_{2k-1}cos(2k\pi t) + A_{2k}sin(2k\pi t)) + \Delta Y(t), \qquad (5)$$

where $Y(t)$ is the measurements of $XCO_2$, $XCH_4$ or $XCO$; $A_0$ is the measurements (backgrounds), and $A_1$ - $A_6$ are the amplitudes of the periodic variations during the year (seasonal variation); $\Delta Y(t)$ is the measurement without background and

seasonal variations, representing the day-to-day variation. Note that, we assume there are no trends of these species due to a relatively short time coverage of about one year. Figure 10 (c, d) show the correlations between the $\Delta XCO$ and $\Delta XCO_2$ daily means and between the $\Delta XCO$ and $\Delta XCH_4$ daily means. The correlation coefficient (R) between XCO and $XCO_2$ increases from 0.50 to 0.66, and the R between XCO and $XCH_4$ increases from 0.67 to 0.82. The seasonal variation of $\Delta XCO_2$ still can be observed, but the amplitude is much reduced. There is almost no seasonal variation of $\Delta XCH_4$. Figure 11 shows the

correlations in each season. The good correlations between $\Delta XCO$ and $\Delta XCH_4$ are found for the whole year, with the R in the range of 0.72-0.87. There is a good correlation (R>=0.85) between $\Delta XCO$ and $\Delta XCO_2$ in autumn and winter, and a worse correlation (R=0.47) in spring and (R=0.57) in summer. It is assumed that the random distribution of the $\Delta XCO$ is symmetric, and the lowest $\Delta XCO$ is about -36 ppb. Therefore, each day with a $\Delta XCO > 36$ ppb is classified as a polluted day, vice versa. In total, we have 28 polluted days and 187 clean days. FTIR measurements show the $\Delta XCO$, $\Delta XCO_2$ and

$\Delta XCH_4$ are much larger in the polluted days than those in the clean days (see Table 3).

The 10-days backward trajectories for polluted and clean days classified by CO measurements are also plotted using the Lagrangian particle dispersion model version 9.02 (FLEXPART). The FLEXPART is able to simulate a large range of atmospheric transport processes, taking mean flow, deep convection, and turbulence into account. The backward running of FLEXPART provides the release-receptor relationship, which is applied to study the source and transport of the observations from a mea-

surement site. In this study, 20000 air particles are released at Xianghe between 10:00 - 14:00 (local time) for days when FTIR measurements are available in the vertical range of surface-2 km, and a 4-D response function to emission inventory is calculated. The model was driven by the meteorological data from the European Centre for Medium Range Weather Forecast (ECMWF). The residence time of particles in output grid cells describes the sensitivity of the receptor to the source. Figure 12 shows the mean air sources for polluted and clean days. It is found that the air is mainly from the south and the local polluted

region (North China) for the polluted days, and is mainly from the north and remote clean places (Inner Mongolia, Mongolia and Russia) for the clean days.

## 4  Satellite validation

### 4.1  Methodology

In this section, the FTIR $XCO_2$, $XCH_4$ and XCO measurements at Xianghe are used to compare with the OCO-2 $XCO_2$ and

TROPOMI $XCH_4$ and XCO satellite observations. There are 215 days' measurements of 15435 individual FTIR retrievals. The co-located FTIR-satellite data pairs are selected based on spatial-temporal collocation criteria. The detailed selection criteria

for each target (OCO-2 $XCO_2$, TROPOMI $XCH_4$ and TROPOMI XCO) are described in the subsections 4.2 and 4.3: they account for the scan width of the satellite instrument and the characteristics of the target species.

According to Rodgers and Connor (2003), the differences in a priori profiles should be taken into account when comparing ground-based FTIR and satellite observations. TCCON $CO_2$, $CH_4$ and CO a priori profiles (70 layers) have been discussed in Section 3.1. OCO-2 $CO_2$ a priori profiles (19 layers) are created based on the GLOBALVIEW dataset and change with time and location (O'Dell et al., 2012). TROPOMI uses the global chemical transport model TM5 to get $CH_4$ and CO a priori profiles (12 layers) (Borsdorff et al., 2018; Hasekamp et al., 2019). The TM5 model data are monthly means with a horizontal resolution of 2°latitude ×3°longitude and 60 vertical levels (Krol et al., 2005). In this study, the satellite a priori profile (OCO-2 or TROPOMI) is taken to be the common a priori profile in the comparison. To substitute the satellite a priori profile in the FTIR retrieval we follow Rodgers and Connor (2003):

$$X'_{FTIR} = X_{FTIR} + (\boldsymbol{A} - \boldsymbol{I})(\boldsymbol{x}_{a,FTIR} - \boldsymbol{x}_{a,SAT}), \tag{6}$$

where $X'_{FTIR}$ is the FTIR retrieved total column using the satellite a priori profile, $X_{FTIR}$ is the original FTIR retrieval, $\boldsymbol{A}$ is the FTIR TCCON column averaging kernel, $\boldsymbol{I}$ is the unit vector, $\boldsymbol{x}_{a,FTIR}$ and $\boldsymbol{x}_{a,SAT}$ are the a priori partial column profiles of FTIR and satellite retrievals, respectively. As the vertical layering of the FTIR retrieval is different from that of the satellite retrieval (OCO-2 or TROPOMI), the satellite a priori profile is re-gridded to the FTIR layer. After re-gridding, the total a priori column remains unchanged (Langerock et al., 2015).

To compare the FTIR and satellite column measurements, the satellite measurements are corrected for a possible difference between the altitudes of its ground pixel and that of the FTIR site at Xianghe. If the surface altitude of the satellite footprint is higher than the altitude of the FTIR instrument, the FTIR a priori profile ($\boldsymbol{x}_{a,FTIR}$) is used to fill the gap between the satellite lowest level ($P_{s,SAT}$) and the FTIR height ($P_{s,FTIR}$), otherwise the satellite a priori profile is considered to be the profile between the satellite lowest level and the FTIR height. Then the partial column of dry air ($PC_{dry,air}$) or target species ($PC_{gas}$) between the satellite footprint surface altitude and the FTIR surface altitude is calculated as

$$PC_{dry,air} = \int_{P_{s,SAT}}^{P_{s,FTIR}} \frac{dP}{g_{(P)}(m_{dry,air} + m_{h2o}\overline{\nu}_{h2o})}, \tag{7}$$

$$PC_{gas} = \int_{P_{s,SAT}}^{P_{s,FTIR}} \frac{dP}{g_{(P)}(m_{dry,air} + m_{h2o}\overline{\nu}_{h2o})} x_{(P)}, \tag{8}$$

where $g_{(P)}$ is gravitational acceleration at height $P$, $x_{(P)}$ is the a priori VMR profile of each target gas, $\overline{\nu}_{h2o}$ is the VMR of water vapor in the dry air, calculated as

$$\overline{\nu}_{h2o} = \frac{\nu_{h2o}}{1 - \nu_{h2o}}, \tag{9}$$

where $\nu_{h2o}$ is the VMR of water vapor in the wet air. Then each satellite pixel measurement is scaled with one scaling factor ($\alpha$) related to satellite pixel level, which is computed as

$$\alpha = \frac{(TC_{gas}^{SAT} + PC_{gas})}{TC_{gas}^{SAT}} \left/ \frac{(TC_{dry,air}^{SAT} + PC_{dry,air})}{TC_{dry,air}^{SAT}} \right., \tag{10}$$

where $TC_{dry,air}^{SAT}$ and $TC_{gas}^{SAT}$ are the total column of dry air and target species in the satellite measurement column. The random error of FTIR measurements together with the systematic and random errors of satellite measurements are considered here for the comparison.

## 4.2 OCO-2

OCO-2 incorporates three imaging grating spectrometers to measure near-infrared spectra. The spectral resolution of OCO-2 is approximately 20 times lower than that of the TCCON FTIR (0.02 cm$^{-1}$) instruments (Frankenberg et al., 2015). OCO-2 collects 8 soundings over its 0.8°swath width every 0.333s with a 16-day repeat cycle (https://ocov2.jpl.nasa.gov/observatory/instrument/). The OCO-2 XCO$_2$ measurements are retrieved by the ACOS retrieval algorithm (O'Dell et al., 2012), based on the optimal estimation method. Three bands (0.756 μm, 1.61μm and 2.06 μm) are used in the XCO$_2$ retrieval. The a priori surface

pressure, profiles of temperature and water vapor are from 3-hourly ECMWF model forecast fields and linearly interpolated in space and time to the satellite footprint. Note that there are three versions (v7, v8 and v9) available on the NASA website for the OCO-2 data. Each version comes in two variants: full and lite. The full variant contains all the retrieved parameters, but without any post-correction applied to the data. The lite variant only includes some important parameters, but the data are corrected in terms of a footprint-dependent bias, a parameter-dependent bias and a scaling bias according to the WMO

trace-gas standard scale. Compared to v7, many parameters have been improved in v8, such as latitude-dependent problems, surface model, spectroscopy, potential instrumental problems, atmospheric scattering by clouds and aerosols, a spatial-temporal sampling error of a priori surface pressure and the systematic pointing offsets (O'Dell et al., 2012). Based on v8, v9 has a better estimation of the surface pressure, and it shows a better performance in regions with rough topography such as over Lauder (New Zealand) (Kiel et al., 2019). In this study, the latest v9 lite data are selected (https://ocov2.jpl.nasa.gov).

The satellite measurements are selected within 5° latitude × 10° longitude around Xianghe, these are the same criteria as adopted by Wunch et al. (2017). For each FTIR measurement, the nearby satellite measurement in the spatial collocation box, with less than 2-hours' measurement time difference, is chosen to form one FTIR-satellite data pair. Note that there are nadir and glint observational modes of OCO-2 measurements over Xianghe, and these two types of measurements are combined together to get a statistically robust result with 28 data pairs (Wunch et al., 2017).

The time series of the co-located OCO-2 and ground-based FTIR data from 27 June 2018 to 31 May 2019 (last date of satellite data availability) is shown in Figure 13. To avoid the influence from the cloud, we select the co-located data pair, which has at least 20 OCO-2 measurements within the box. The upper panel in Figure 13 (left) shows the daily mean bias of measured XCO$_2$ from OCO-2 and FTIR. The mean of OCO-2 measurements is 0.62 ppm lower than that of the FTIR measurements, with a STD of 1.20 ppm. The absolute differences between OCO-2 v9 lite data and Xianghe FTIR data are

comparable with the results found for the v7 lite products in Wunch et al. (2017) for other TCCON stations with biases ranging from $-0.7 \pm 1.32$ ppm (Wollongong) to $0.9 \pm 1.49$ ppm (Karlsruhe) in land glint mode and ranging from $-0.1 \pm 1.04$ ppm (Wollongong) to $1.6 \pm 2.05$ ppm (Garmisch) in nadir mode. The scatter plot of OCO-2 and FTIR at Xianghe is shown in the right panel in Figure 13: the derived correlation coefficient (R) is 0.959. We can conclude that OCO-2 data are in good

agreement with the Xianghe FTIR data, and in particular, that OCO-2 captures the seasonal cycle of $XCO_2$ at Xianghe, with a maximum in winter-spring and a minimum in late summer.

## 4.3 TROPOMI

In this section, the TROPOMI $XCH_4$ and XCO are compared with the FTIR measurements at Xianghe. TROPOMI is a grating spectrometer measuring solar radiation reflected by the Earth and observes in the ultraviolet and visible, near-infrared and shortwave infrared spectral regions. It has a wide swath of around 2600 km across the track and a daily global coverage of the Earth. The spatial resolution of TROPOMI is about 7km × 7km before 6 August 2019 and then it changes to 7.2km × 5.6km. The TROPOMI CO data are provided in three different data streams: the near-real-time (NRTI) stream (since June 2018, with the same starting month as FTIR measurements at Xianghe), the Offline stream (OFFL) and the Reprocessing (RPRO) stream (Landgraf et al., 2019a). $CH_4$ data are provided in bias-corrected and not-corrected versions (Landgraf et al., 2019b). In our validation, we considered the off-line and reprocessed CO data, from processor versions 01.02 and higher. For $CH_4$, we also look at bias-corrected data with processor versions of 01.02 and higher. The FTIR measurements at Xianghe can provide very useful information in a polluted area of North China, which is also important for TROPOMI product validation.

TROPOMI uses the RemoTeC algorithm to retrieve $CH_4$ column using the 0.757-0.774 μm $O_2$ absorption band and 2.305-2.385 μm $CH_4$ absorption band (Hasekamp et al., 2019). The requirements for the accuracy and precision for TROPOMI $XCH_4$ are 1% and 1.5%, respectively (Hasekamp et al., 2019). We select TROPOMI $XCH_4$ measurements that occur within 1 hour of FTIR measurements and within a distance of 100 km from the Xianghe station based on the collocation criteria adopted at other TCCON sites (Lambert et al., 2019). In agreement with Landgraf et al. (2019b), the TROPOMI pixels are selected with a quality assurance value above 0.5, which removes pixels with processing errors, anomalously high signals and increasing specular reflection of sunlight by the sea surface (Hasekamp et al., 2019). Similar to OCO-2, to reduce the influence from the clouds, we only select the days when there are at least 5 co-located TROPOMI $CH_4$ pixels.

The left panel in Figure 14 shows the time series of co-located TROPOMI and FTIR $XCH_4$ daily means and their relative biases (%,(satellite-FTIR)/FTIR) from 27 June 2018 to 19 July 2019 (86 days). The co-located TROPOMI and FTIR $XCH_4$ data pairs at Xianghe are distributed evenly in all seasons. The mean bias is -0.60 %, which is within the S5P validation requirement of a bias of 1%. In addition, the STD of the relative biases is 0.55 %, which also meets the S5P mission requirement of 1.5% (Lambert et al., 2019). The R between TROPOMI and FTIR $XCH_4$ daily means is 0.834 (Figure 14, right panel). According to the TROPOMI validation report (Lambert et al., 2019), the bias at Xianghe is comparable to the ones at Tsukuba, Lamont and Rikubetsu (similar latitude band).

The TROPOMI XCO measurements are retrieved from the SICOR algorithm (Hasekamp et al., 2019) in the 2.3 μm spectral range. The retrieved TROPOMI CO data is in the unit of total column density (molecules/cm$^2$), so we converted them to to XCO (ppb) values for comparison with FTIR XCO measurements (Langerock et al., 2015):

$$XCO = \frac{TC_{CO}}{TC_{dry,air}^{SAT}}, \tag{11}$$

where XCO is the total column-averaged dry-air mole fraction of TROPOMI CO measurements. $TC_{CO}$ is the total column density of TROPOMI CO measurements, $TC_{dry,air}^{SAT}$ is total column density of dry air in the satellite measurement column. Because CO is relatively reactive compared to $CH_4$, we must reduce the measurement time and location differences in the colocation criteria. Therefore, the TROPOMI observations are selected within 30 minutes of each FTIR measurement and within a maximum distance of 50 km away from the FTIR site and along the light path of the ground-based FTIR measurements. Similar to $CH_4$, we only select the days when there are at least 5 co-located TROPOMI CO pixels. In addition, to reduce the impact from long light paths through the atmosphere (Landgraf et al., 2019a), the TROPOMI measurements with a SZA larger than 80°or a satellite zenith angle larger than 65°are filtered out. And we only select the TROPOMI CO products in clear sky cases with cloud height below 500 m and cloud optical depth < 0.5.

The left panel of Figure 15 shows the time series and relative biases of co-located TROPOMI and FTIR XCO daily means at Xianghe from 27 June 2018 to 31 May 2019 (70 days). In addition, the co-located TROPOMI and FTIR XCO data pairs at Xianghe are also distributed evenly in all seasons. The mean bias and STD between TROPOMI and FTIR are 2.05% and 7.82%, respectively, which are within the S5P mission requirement (bias < 15% and STD < 10%). Compared to other TCCON sites(Lambert et al., 2019), the mean relative bias is relatively low. The good agreement between TROPOMI and FTIR XCO with a R of 0.961 (Figure 15, right panel) highlights the good performance of TROPOMI over Xianghe.

## 5   Conclusions

A new ground-based FTIR Bruker 125HR instrument has been in operation since 14 June 2018 at Xianghe (39.75 °N, 116.96 °E) in North China. It performs NIR solar absorption measurements of $XCO_2$, $XCH_4$ and XCO following the TCCON operation and data analysis procedures since June 2018. Regular HCl cell measurements show that the ME loss is within 5% for the whole time period, and the PE deviation remains within 0.02 rad in most time, confirming that the ILS of the FTIR spectrometer is stable during the considered time period and meets the TCCON requirements. $XCO_2$, $XCH_4$ and XCO have been retrieved using the TCCON standard algorithm GGG2014.

Because the InGaAs spectra have been recorded in AC mode before 31 May 2019 and AC+DC mode afterwards, we designed a filtering method based on SNR and SI. Application of this filtering to the spectra before 31 May 2019 shows that about 85% of the spectra have the required quality. The thus achieved precision of 0.8 ppm for the retrieved $XCO_2$ complies with the TCCON requirements, demonstrating that the SI filtering can overcome the absence of the DC signal in the period before end of May 2019.

During this 1-year period of measurements, a clear seasonal variation of $XCO_2$ has been observed, with lowest values of $401.58 \pm 1.32$ ppm in summer and highest values of $414.27 \pm 0.94$ ppm in winter. Low $XCH_4$ concentrations are observed in spring ($1.857 \pm 0.014$ ppm) and high values in autumn and summer ($1.893 \pm 0.017$ ppm). For XCO there is no clear seasonal variation, but a large day-to-day variability. According to the FTIR measurements, it is found that the high values of $XCO_2$, $XCH_4$ and XCO are highly related, which is affected by the large local anthropogenic emissions.

Comparisons between the $XCO_2$, $XCH_4$ and XCO FTIR measurements at Xianghe and satellite data are shown to illustrate that this site is an interesting additional site for satellite validation in northern China. The mean bias between FTIR and OCO-2 $XCO_2$ measurements is -0.62 ppm with a STD of 1.20 ppm. The mean and STD of the relative differences between FTIR and TROPOMI $XCH_4$ measurements are -0.60% and 0.55%, respectively. Both results are consistent with comparisons between these satellite data and other TCCON sites. The mean and STD of the relative differences between FTIR and TROPOMI XCO measurements are 2.05% and 7.82%, respectively, which is within the S5P mission requirements. However, the mean relative bias is lower than what is observed at other TCCON sites. This is due to the fact that Xianghe is a highly polluted site while the TROPOMI a priori CO profiles underestimate CO in the lower polluted troposphere. The variability is however well captured by TROPOMI. Therefore, the ground-based FTIR measurements at Xianghe appear very useful to evaluate the performance of the greenhouse gases observing satellites (OCO-2 and TROPOMI) above this region.

In summary, this study shows that the ground-based FTIR Xianghe data for $XCO_2$, $XCH_4$ and XCO comply with the TCCON specifications. The objective is to become a TCCON affiliated site in future. As Xianghe is a rather polluted location, it can provide useful information for the study of the carbon cycle in North China, and for the validation of satellite observations over this region.

## 6   Data availability

The Xianghe FTIR $CO_2$, $CH_4$ and CO data can be accessed at https://doi.org/10.18758/71021049 (Yang et al., 2019). The OCO-2 data are publicly available (https://ocov2.jpl.nasa.gov). The TROPOMI data are publicly available (https://scihub.copernicus.eu/).

*Author contributions.*  MZ, PW, GW and MDM designed the experiment. YY, MZ, CH,TW, DJ, CV, NK operated and maintained the FTIR instruments. YY, MZ, BL and MKS performed the satellite validation. YY and MZ wrote the manuscript and all authors read and provided comments on the paper.

*Competing interests.*  The authors declare that they have no conflict of interest.

*Acknowledgements.*  The work is supported by National Key R & D Program of China (Nos. 2017YFB0504000 and 2017YFC1501701), the National Natural Science Foundation of China (No.41575034) and China Scholarship Council. We want to thank TCCON community for sharing the retrieval code GGG2014. We also want to thank Weidong Nan, Qun Cheng and Qing Yao at Xianghe site, Rongshi Zou (IAP) and Francis Scolas for the FTIR maintenance.

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

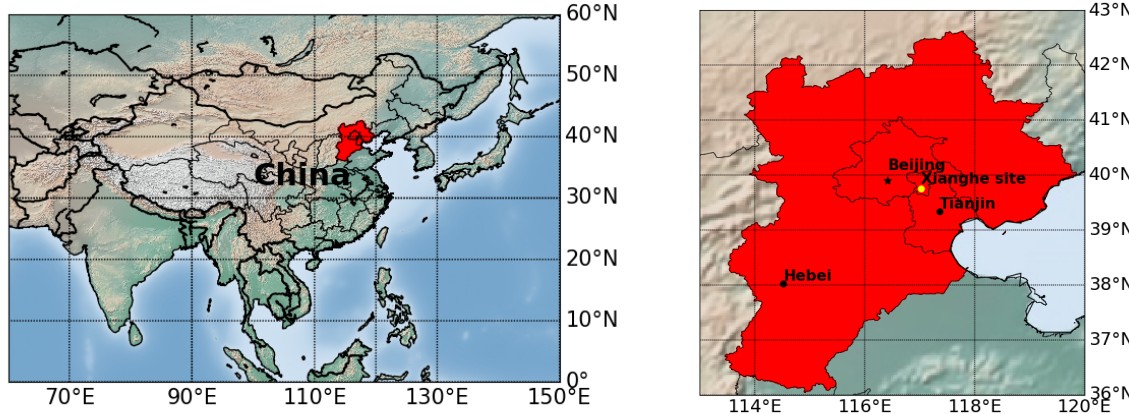

**Figure 1.** Map of China, the Beijing-Tianjin-Hebei region is the red shadow area which is also zoomed in the right panel. It is one of the most populous and economically dynamic regions in China. Xianghe site (yellow spot in the right panel) is located in Xianghe county, about 50 km to the east-southeast of Beijing and 70 km to the north-northwest of Tianjin, acting an integrated transportation and transfer center in this region.

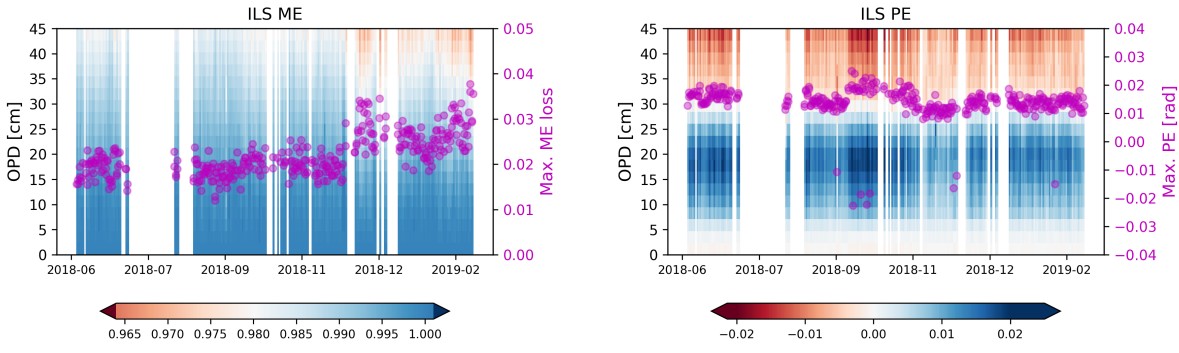

**Figure 2.** The modulation efficiency (ME, left panel) and phase error (PE, right panel) along the optical path difference (OPD) at Xianghe. The purple dots are the maximum ME loss (left) and the maximum PE deviation (right).

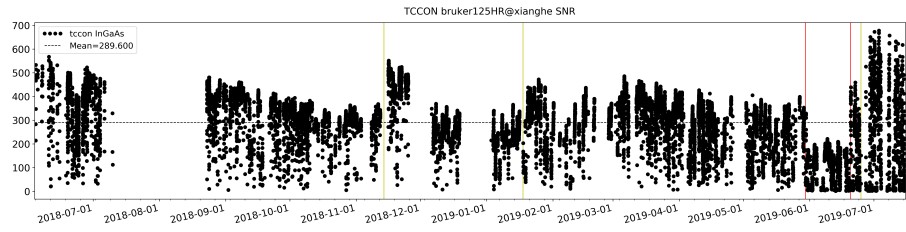

**Figure 3.** Time series of the signal-to-noise ratio (SNR) of the InGaAs spectra from the FTIR in Xianghe. The yellow lines indicate solar tracker maintenances: cleaning of the mirrors on 14 November 2018 and 25 June 2019, and replacement of the degraded mirror on February 1, 2019. The first red line indicates the day when we add DC signal recording (31 May 2019) and the second red line indicates the day when the aperture was increased from 0.5mm to 0.8mm (19 June 2019).

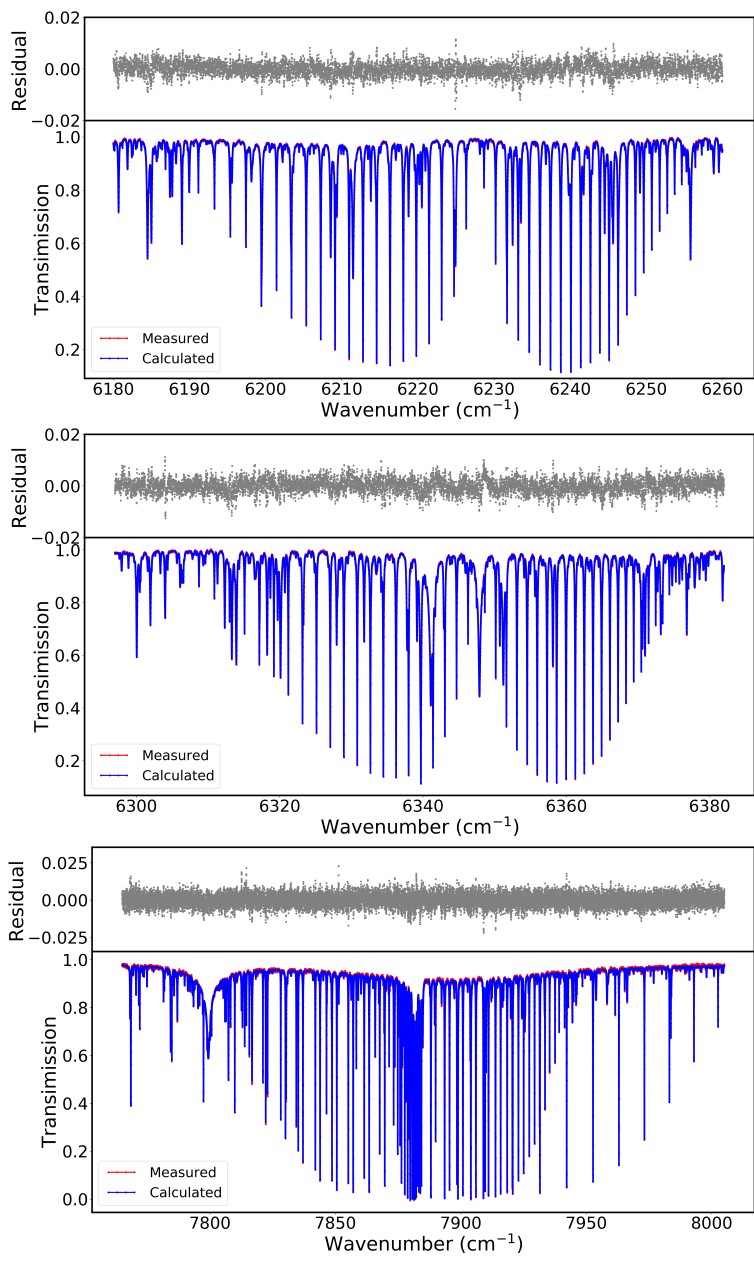

**Figure 4.** Measured and calculated spectrum of $CO_2$ in spectral window of 6180-6260 $cm^{-1}$ (upper panel), $CO_2$ in spectral window of 6297-6382 $cm^{-1}$ (middle panel) and $O_2$ in spectral window of 7765-8005 $cm^{-1}$ (bottom panel).

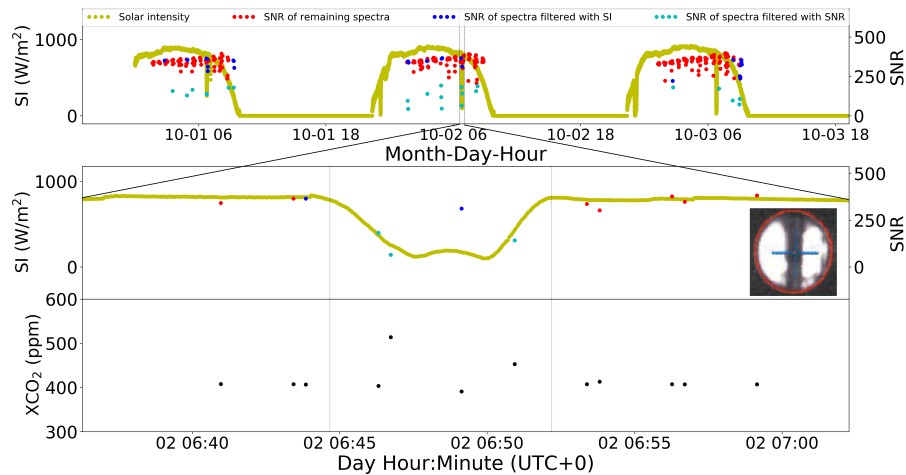

**Figure 5.** The solar direct intensity from 1 October 2018 to 3 October 2018, with a zoom on the time period between 6:36 and 7:02 (UTC) on 2 October 2018 when the tower shadow is observed by the FTIR (see the inside camera image, middle panel). The blue dots in the upper panel denote the SNR of spectra filtered with solar intensity (SI), the cyan dots denote the SNR of spectra filtered with SNR, and the red dots denote the SNR of spectra after both filtering. The affected $XCO_2$ retrieval during the shadow appearing time are displayed in the bottom panel.

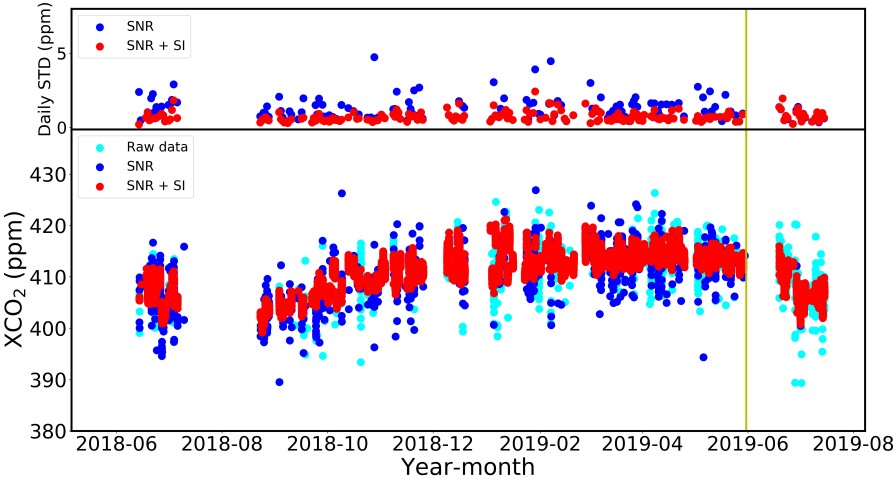

**Figure 6.** Time series of $XCO_2$ from 14 June 2018 to 19 July 2019 before and after spectra selection. The cyan dots denote all the raw data, the blue dots in bottom panel are those retrieved $XCO_2$ after SNR filtering and the red dots are those retrieved $XCO_2$ after SNR + SI filtering. In upper panel, the blue dots denote the daily standard deviation of $XCO_2$ only with SNR filtering while the red ones denote those with SNR + SI filtering. The yellow line denotes the day (31 May 2019) when we added the DC signal recording.

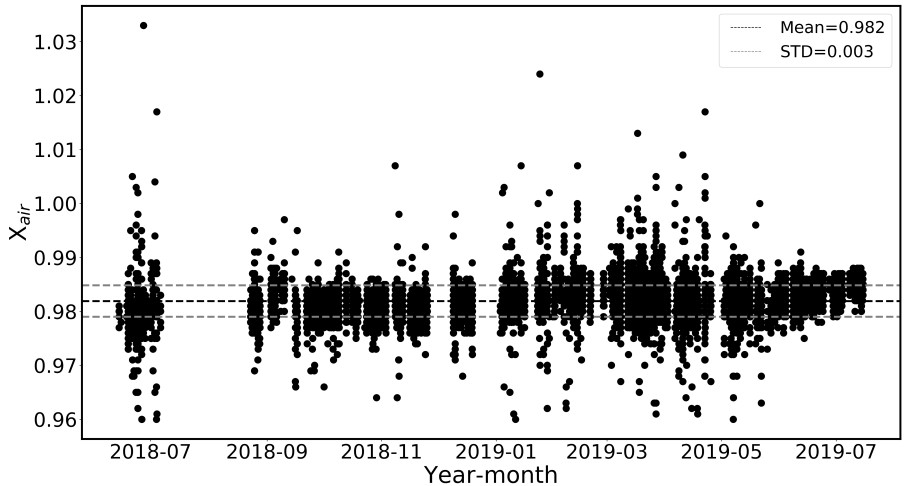

**Figure 7.** Time series of $X_{air}$.

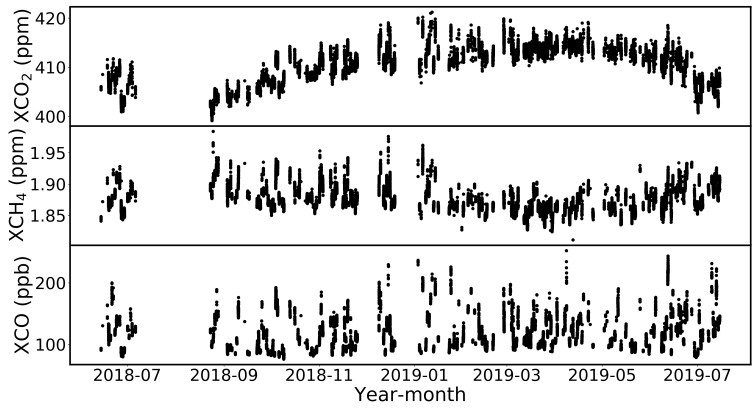

**Figure 8.** Time series of $XCO_2$, $XCH_4$ and XCO covering the period from 14 June 2018 to 19 July 2019.

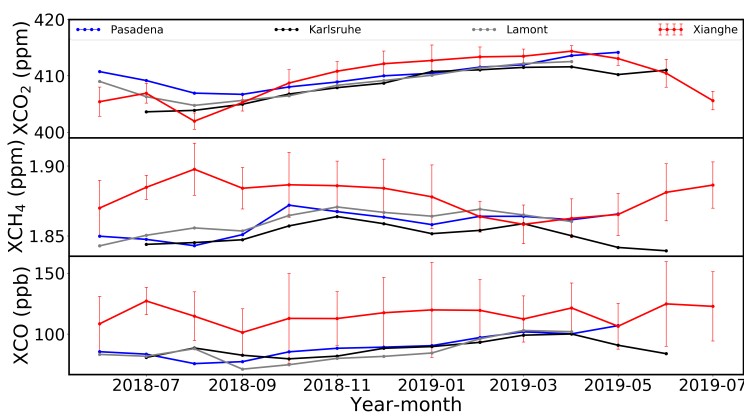

**Figure 9.** Monthly mean of $XCO_2$, $XCH_4$ and XCO at Pasadena, Karlsruhe, Lamont and Xianghe. The error bars are the monthly STDs of $XCO_2$, $XCH_4$ and XCO at Xianghe.

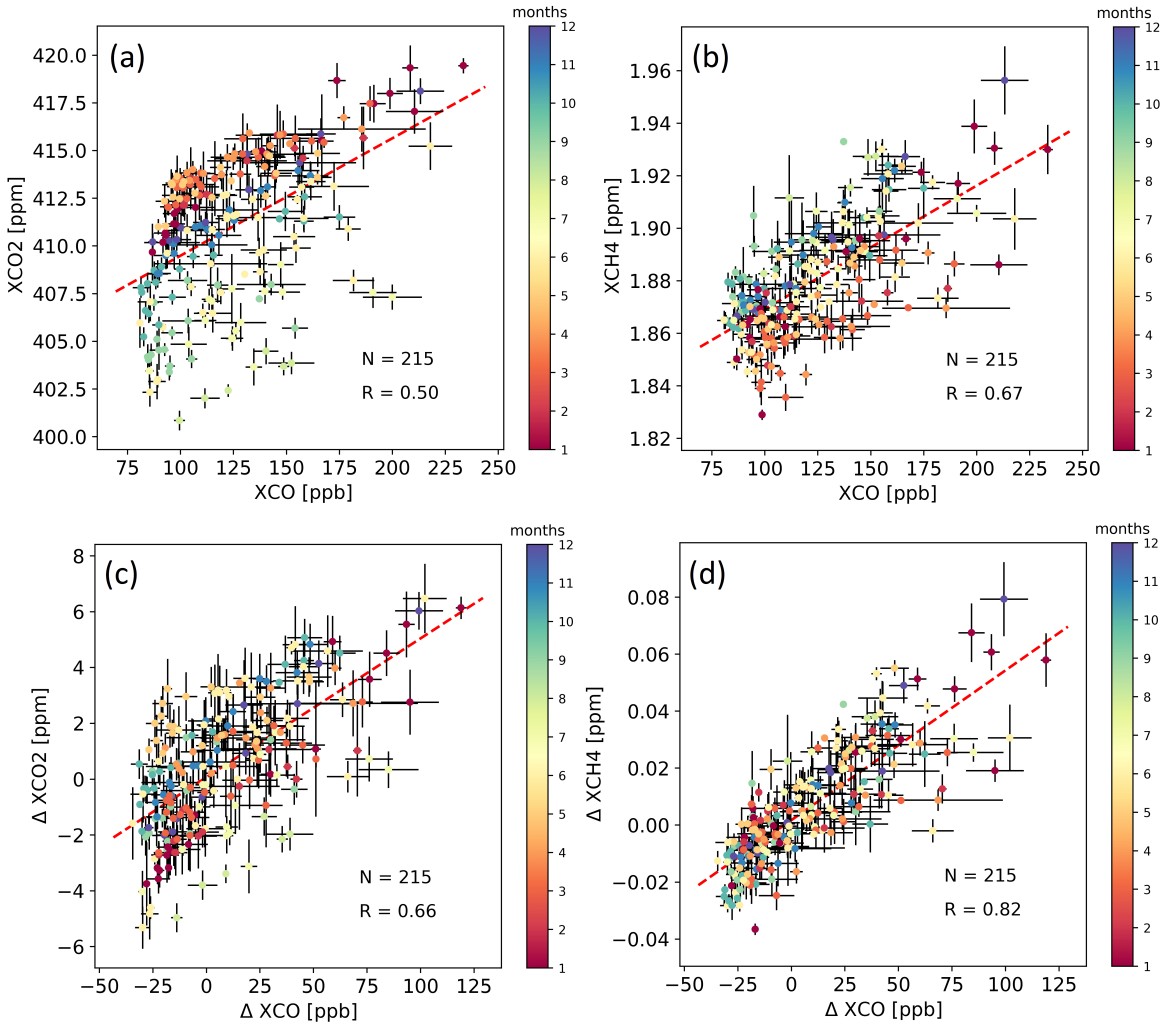

**Figure 10.** The correlation plots between the XCO and $XCO_2$ and $XCH_4$ daily means (a, b) and the correlation plots between the $\Delta XCO$ and $\Delta XCO_2$, and $\Delta XCH_4$ daily means (c, d) from FTIR TCCON-type measurements at Xianghe. The dash red line is the linear fit. The N is the number of the measurement days, and R is the correlation coefficient. The error bar is the standard deviation of the measurements in each day. The data are colored with the measurement months.

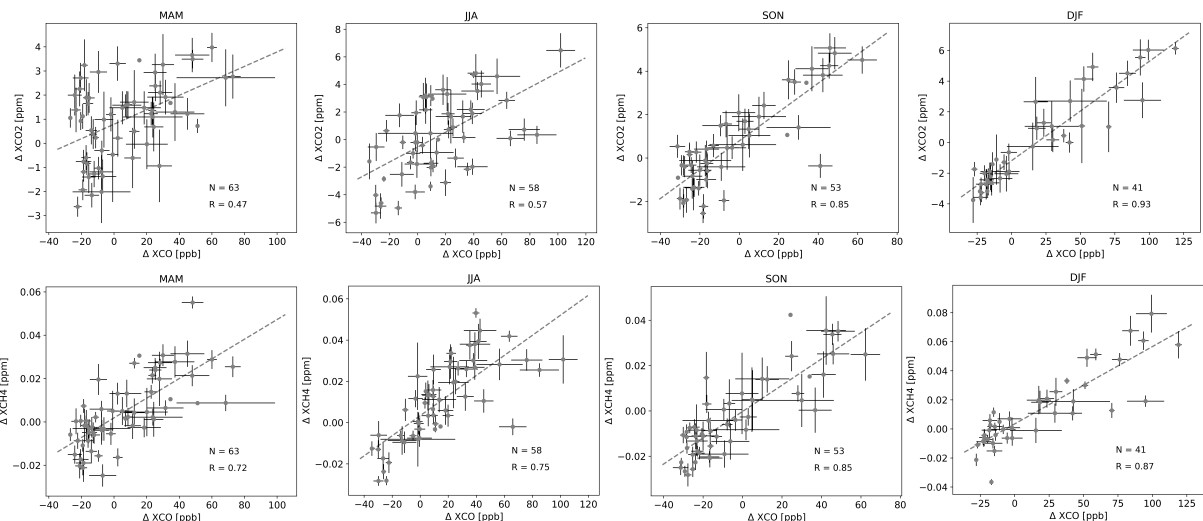

**Figure 11.** Upper panels: the correlation plots between the $\Delta XCO$ and $\Delta XCO_2$ in four seasons (spring: March, April, May (MAM); summer: June, July, August (JJA); autumn: September, October, November (SON); winter: December, January, February (DJF)). Lower panels: the correlation plots between the $\Delta XCO$ and $\Delta XCH_4$ in four seasons. The dash line is the linear fit. The N is the number of the measurement days, and R is the correlation coefficient. The error bar is the standard deviation of the measurements in each day.

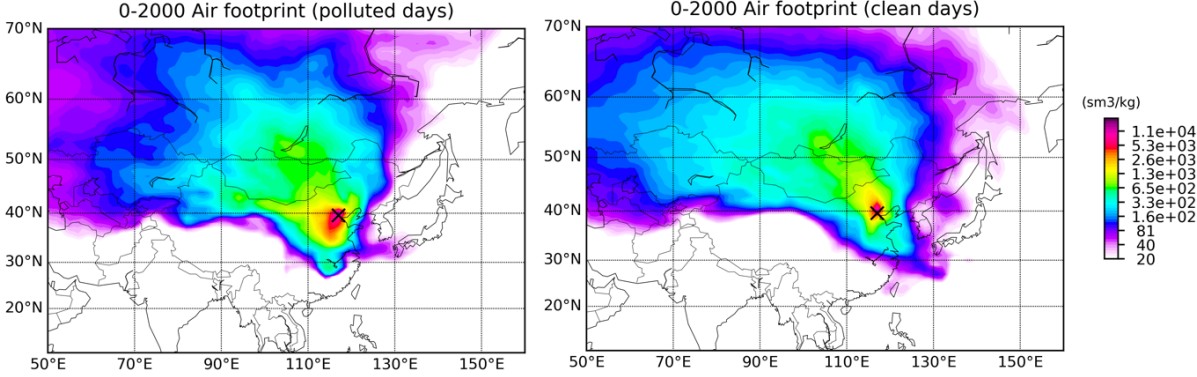

**Figure 12.** The mean emission response sensitivities of the air mass at Xianghe (the cross symbol) for polluted (left) and clean (right) days in the vertical range from surface to 2000 m a.s.l. simulated with a 10 day backward run with FLEXPART v9.02.

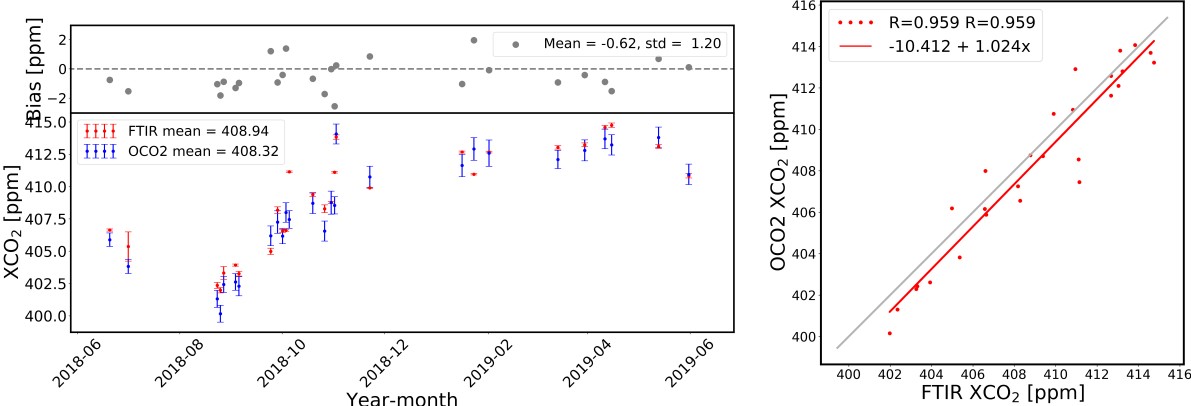

**Figure 13.** Left panel: Time series of daily mean co-located OCO-2 and ground-based FTIR XCO$_2$ data at Xianghe (lower plot) and the bias between them (upper plot). Right panel: Correlation plot between co-located daily mean XCO$_2$ data from OCO-2 and FTIR at Xianghe.

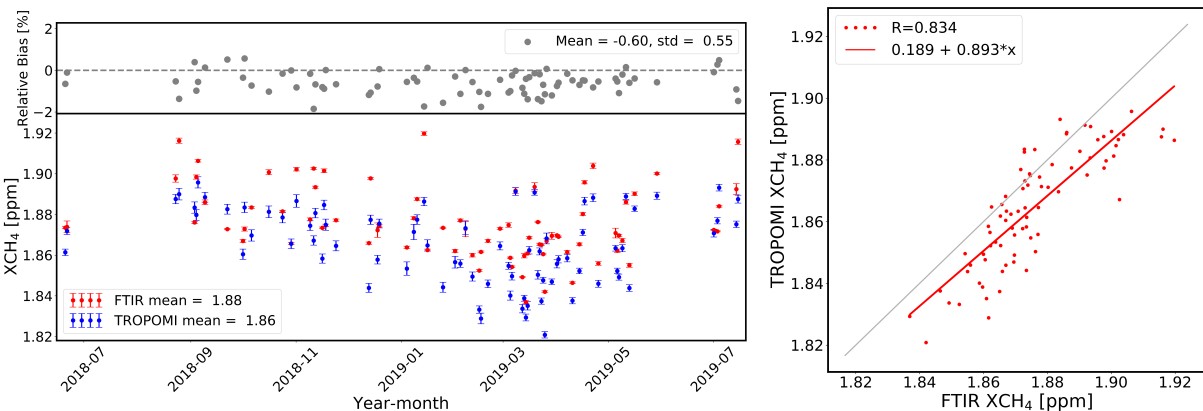

**Figure 14.** Left panel: Time series of daily mean co-located TROPOMI and ground-based FTIR XCH$_4$ data at Xianghe (lower plot) and the bias between them (upper plot). Right panel: Correlation plot between co-located daily mean XCH$_4$ data from TROPOMI and FTIR at Xianghe.

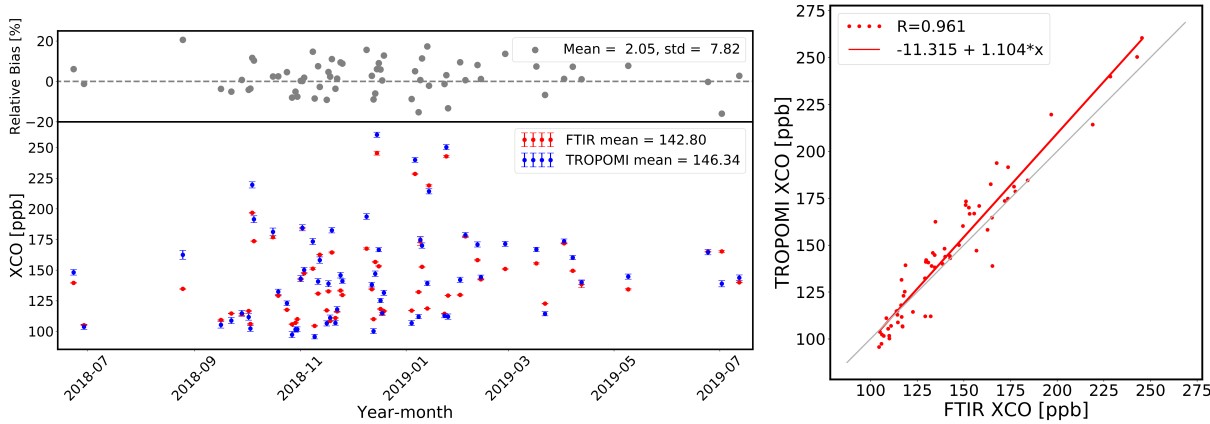

**Figure 15.** Left panel: Time series of daily mean co-located TROPOMI and ground-based FTIR XCO data at Xianghe (lower plot) and the bias between them (upper plot). Right panel: Correlation plot between co-located daily mean XCO data from TROPOMI and FTIR at Xianghe.

**Table 1.** The standard deviation (STD) of XCO$_2$, XCH$_4$ and XCO of the remaining data after each solar intensity (SI) filtering for each day and the percentage of the remaining spectra amount from 14 June 2018 to 31 December 2018. Only measurements with SZA less than 30°are selected.

| $\beta$ (%) | $\gamma$ (%) | STD(XCO$_2$) (ppm) | STD(XCH$_4$) (ppm) | STD(XCO) (ppb) | Percentage of remaining spectra (%) |
|---|---|---|---|---|---|
| 85 | 10 | 0.749 | 0.0042 | 3.032 | 89.5 |
| 85 | 5 | 0.723 | 0.0036 | 2.954 | 88.9 |
| 85 | 0 | 0.667 | 0.0035 | 2.997 | 88.1 |
| 90 | 10 | 0.651 | 0.0036 | 2.904 | 86.6 |
| 90 | 5 | 0.648 | 0.0035 | 2.908 | 85.8 |
| 90 | 0 | 0.496 | 0.0035 | 2.559 | 84.7 |
| 95 | 10 | 0.450 | 0.0029 | 2.421 | 78.5 |
| 95 | 5 | 0.447 | 0.0029 | 2.436 | 77.1 |
| 95 | 0 | 0.423 | 0.0026 | 2.438 | 75.2 |

**Table 2.** Average XCO$_2$ daily standard deviation after (a) no filtering, (b) SNR filtering, (c) both SNR filtering and SI filtering during the AC mode and AC+DC mode periods. The unit is ppm. Only those measurements with SZA less than 30°are considered.

|   | AC mode | AC+DC mode |
|---|---|---|
| a | 1.19 | 1.06 |
| b | 0.94 | 0.53 |
| c | 0.57 | 0.88 |

**Table 3.** The mean and standard deviation of $\Delta XCO$, $\Delta XCO_2$ and $\Delta XCH_4$ at polluted and clean days.

|   | Polluted days | Clean days |
|---|---|---|
| $\Delta XCO$ [ppb] | -9.51 $\pm$ 21.10 | 58.40 $\pm$ 19.58 |
| $\Delta XCO_2$ [ppm] | -0.64 $\pm$ 2.05 | 2.75 $\pm$ 2.01 |
| $\Delta XCH_4$ [ppm] | -0.003 $\pm$ 0.016 | 0.029 $\pm$ 0.019 |