# Peer review of "New ground-based Fourirer-transform near-infrared solar absorption measurements of XCO2, XCH4 and XCO at Xianghe, China"

_Earth System Science Data, 2019_

## Referee Comment (RC1) · Anonymous Referee #1 · 16 Dec 2019

General comments: The paper describes a new ground-based FTIR measurement site in China, presents a study of greenhouse gases using a ground-based Fourier Transform Infrared Spectrometer of the Bruker IFS 125HR. The measured spectra are analyzed using the GFIT2014 code and the retrieved Xgas are presented. The XCO2 retrieved from the ground-based FTIR are compared to XCO2 retrieved from OCO-2, XCH4 and XCO retrieved from the ground-based FTIR are compared to the XCH4 and XCO retrieved from TROPOMI satellite observations. However, the description of the paper lacks scientific significance and originality. Also, the time series of target gases cover only one year period, so some discussion and conclusions are not representative.

[Figure]

Specific comments: 1. The aim of the study is to validate satellite data using the FTIR observations, but the paper doesn't describe how to evaluate the accuracy or precision of the FTIR observations. 2. The discussion about day to day variations of Xgas in section 3.3 only use 6-day data, for example in Fig. 10 and 11, so the conclusions about the day to day variation trend of Xgas and the emission source are not reliable and representative. 3. The FTIR measurements need to be very precise and accurate to be useful for satellite validation or model studies, a proper demonstration over a longer period of time is therefore needed for the site. However, the data cover only one year period. 4. In Line 25 Page 11, "The retrieved TROPOMI CO data is in the unit of total column density (molecules/cm2), so we converted them to to XCO (ppb) values for comparison with FTIR XCO measurements", there should be a short description of the method and cite a reference. 5. In Line 12 Page 12, "Regular HCl cell measurements show that the ME loss is within 2% and the PE remains within 0.02 rad", the conclusion is not consistent with the ME results in Fig. 2. 6. There are no unit for the mean and std value in the Fig. 12, Fig .13 and Fig. 14. 7. In Line 14 Page 1, "The rapid economic growth of China has contributed to 30% of the global total carbon dioxide (CO2) emissions from fossil fuel consumption and cement production (Jackson et al., 2017)", the exact contribution is about 28.5% according to the results in Jackson et al., 2017.

Please also note the supplement to this comment:
https://www.earth-syst-sci-data-discuss.net/essd-2019-172/essd-2019-172-RC1-supplement.pdf
* * *

---

## Referee Comment (RC2) · Anonymous Referee #2 · 30 Dec 2019

**General comments**

The manuscript "A new site: ground-based FTIR XCO2, XCH4 and XCO measurements at Xianghe, China" by Yang et. al. describes a data set of total column dry air mole fractions of carbon dioxide, methane and carbon monoxide derived from near infrared, solar absorption Fourier transform spectroscopy using methods similar to those of the Total Carbon Column Observing Network (TCCON). The described dataset covers a year of measurements and therefore captures a single seasonal cycle.

The manuscript follows a logical narrative, but in some areas would benefit from additional copy editing.

[Figure]

Data of this type, from an urban area of China, are likely to be of value to the satellite and model validation communities and I would encourage the publication of this manuscript subject to a number of clarifications and modifications as detailed below.

**Specific comments**

Reference is made to TCCON throughout the manuscript, however it is not made explicitly clear that the Xianghe site is not currently affiliated to TCCON. This should be stated from the start including what steps are required for the site to become affiliated with TCCON if that is the objective. Additionally, statements such as "this study shows that the Xianghe data comply with the TCCON specifications" at page 12 line 30 are considered quite strong and possibly misleading as, for example, the TCCON requirement for an in-situ validation is not covered in this work. Care should also be taken when making comparison to the TCCON accuracy/precision requirements, throughout the manuscript reference is made to "the 0.8 ppm (SZA less than 80 °) retrieval accuracy of TCCON XCO2." (e.g. at P4 L23), this is actually the target for site-to-site biases within the TCCON. The actual single instrument/site precision target for TCCON is 0.1% i.e. 0.4 ppm for XCO2.

Precision estimates within the manuscript are based on the standard deviations of daily retrievals of each of the species. This may result in an overly pessimistic estimate as the daily means are potentially influenced by a number of factors such as a diurnal cycle, the influence of local sources in such a heavily urbanised area and a residual airmass dependence of the retrievals as pressure broadening is not accounted for in the GGG2014 spectroscopy. Better and more representative results may be obtained by limiting the windows over which standard deviations are calculated to a smaller range of solar zenith angles or shorter periods.

In sub-section 2.2 it would also be useful to present some example plots of ME and PE as a function of OPD, and as PE can have a maximum value at an OPD other than the maximum, consider presenting a time series of maximum PE. Also, LINEFIT 14.5 can

be used in a number of modes, it should be explicit which mode is used.

The sub-section on signal-to-noise ratio would benefit from a discussion about how SNR is calculated as there are a number of possible methods. Making this explicit would make it easier to make comparisons between sites.

In the paragraph describing satellite missions that can be validated by this type of measurement in the introduction, GOSAT is notable by its absence, and as one of the first satellites dedicated to measuring greenhouse gases probably deserves to be mentioned.

Ongoing measurements of this type are likely to be of interest to a number of users and so a statement about whether future measurements will be added to the referenced dataset, or the TCCON archive if the site should become affiliated to the network, would be useful.

**Technical corrections**

P2 L4-8 Whilst not strictly necessary, it would be useful to provide more context by comparing the current values to their pre-industrial levels and/or stating the current rate of increase.

P2 L10 Provide the radiative forcing of $CO_2$ as a comparison.

P2 L13-14 Try to clarify the distinction that is being made here, i.e. are the two main types of measurement in-situ and remote sensing, or ground-based and satellite?

P3 L1-2 Inclusion of the sentence "To increase the precision of the retrievals, the spectra are cloud filtered based on the separate direct solar irradiation measurements." fragments the description of the manuscript outline, considering removing it.

P3 L33 After "NIR InGaAs" insert spectra or measurements.

P3 L26 - P4 L7 These two paragraphs highlight the importance of consistent representation of scalars and their vectors e.g. m and meters are both used and sometimes there is a space between the scalar and vector and sometimes not.

P5 Eq.1 Appears to have been inserted at the wrong point in the manuscript.

P5 L16 Please give details of these sensors, particularly the type and precision/accuracy of the pressure sensor which is very important for the accuracy of trace gas retrievals.

P5 L20 Does the first "a priori partial column" refer to the tropospheric component? This should be made more explicit.

P6 L10 Consider using affected instead of infected.

P9 Eq.4 The vector **l** is not defined.

P11 L8, L14 and L31 The Landgraf et al references do not contain a year.

Table 2. AC+DC mode column, it would be expected that additional filtering would result in a lower value for SNR+SI (row c) than for SNR (row b) on its own. This doesn't appear to be the case.

---

## Author Comment (AC1) · 7 Feb 2020

Black: referee's comments red: authors' answers First of all, we want to thank the referee for the detailed analysis of our paper. For the details, please look into the paper with keeping track of changes.

Anonymous Referee #1

General comments: The paper describes a new ground-based FTIR measurement site in China, presents a study of greenhouse gases using a ground-based Fourier Transform Infrared Spectrometer of the Bruker IFS 125HR. The measured spectra are analyzed using the GFIT-2014 code and the retrieved Xgas are presented. The XCO2 retrieved from the ground-based FTIR are compared to XCO2 retrieved from OCO-2, XCH4 and XCO retrieved from the ground-based FTIR are compared to the XCH4 and XCO retrieved from TROPOMI satellite observations. However, the description of the paper lacks scientific significance and originality. Also the time series of target gases cover only one year period, so some discussion and conclusions are not representative.

The paper is aim to describe the new ground-based FTIR XCO2, XCH4 and XCO measurements at Xianghe, and to show that the data quality of our FTIR measurement is comply with the TCCON requirement. The site is located in a very polluted area in North China, with no TCCON site at the moment. These measurements are very useful for the climate and air pollution studies as well as related satellite validation.

**Specific comments:**

1. The aim of the study is to validate satellite data using the FTIR observations, but the paper doesn't describe how to evaluate the accuracy or precision of the FTIR observations. In the revised version, we reword the title, abstract, introduction and conclusion to make the target of our paper more clear. In fact, the aim of the study is to describe the ground-based FTIR data at Xianghe, and to show that the data quality of our FTIR measurement is comply with the TCCON requirement.

The data quality at Xianghe is assessed by the instrument line shape, the quality of the spectra, the uncertainty of the metadata, the residual of the fitting, and the standard deviation of the retrievals. The setup of the instrument at Xianghe follows the guidance of the TCCON, and in this paper we prove that all these parameters meet the TCCON requirements. In addition, the spectra at Xianghe are analyzed by the standard retrieval code (GGG2014), where the systematic uncertainty (accuracy) of the retrieval is mainly from the spectroscopy. According to the comparison with aircraft or AirCore measurements (Wunch et al., 2015), the systematic uncertainty of XCO2, XCH4 and XCO retrievals (mainly from the spectroscopy uncertainty). The random uncertainties (precision) of the XCO2, XCH4 and XCO retrievals at Xianghe are evaluated by their standard deviations (see Table 1 and Table 2 in the revised version), and their random uncertainties are all within the TCCON reported errors (Wunch et al., 2015).

2. The discussion about day to day variations of Xgas in section 3.3 only use 6-day data, for example in Fig. 10 and 11, so the conclusions about the day to day variation trend of Xgas and the emission source are not reliable and representative.

Thanks for the suggestions. More analyses are added in the revised to make the statement reliable and representative:

Figure 1 (a, b) shows the correlations between the XCO and XCO2 daily means and between the XCO and XCH4 daily means at Xianghe. XCO2 is high in winter and low in summer, and XCH4 is high in summer and autumn and low in winter. In order to reduce the impact from the seasonal variation, a linear regression model is used to fit the time series of the measurements

 $Y(t) = A_0 + \sum_{k=1}^3 (A_{2k-1} \cos(2k\pi t) + A_{2k} \sin(2k\pi t)) + \Delta Y(t)$ , where Y(t) is the measurements of XCO2, XCH4 or XCO;  $A_0$  the mean of the measurements (background), and  $A_1$ - $A_6$  are the amplitudes of the periodic variations during the year (seasonal variation);  $\Delta Y(t)$  is the measurement without background and seasonal variations, representing the day-to-day variation. Note that, we assume there are no trends of these species due to a relatively short time coverage of about one year. Figure 1 (c, d) show the correlations between the  $\Delta$ XCO and  $\Delta$ XCO2 daily means and between the  $\Delta$ XCO and  $\Delta$ XCH4 daily means. The correlation coefficient (R) between XCO and XCO2 increase from 0.50 to 0.66, and the R between XCO and XCH4 increase from 0.67 to 0.82. The seasonal variation of  $\Delta$ XCO2 still can be observed, but the amplitude is much reduced. There is almost no seasonal variation of  $\Delta$ XCH4. Figure 2 shows the correlations in all seasons. It is found that a good correlation between  $\Delta$ XCO and  $\Delta$ XCH4 is found for the whole year, with R values in the range of 0.72-0.87. There is a good correlation (R>=0.85) between  $\Delta$ XCO and  $\Delta$ XCO2 in autumn and winter, and a slightly weak correlation (R=0.47) in spring and (R=0.57) in summer.

---

## Author Response (AR1)

*Black: referee's comments* *red: authors' answers*
*First of all, we want to thank the referee for the detailed analysis of our paper.*
*For the details, please look into the paper with keeping track of changes.*

Anonymous Referee #1

General comments: The paper describes a new ground-based FTIR measurement site in China, presents a study of greenhouse gases using a ground-based Fourier Transform Infrared Spectrometer of the Bruker IFS 125HR. The measured spectra are analyzed using the GFIT-2014 code and the retrieved Xgas are presented. The XCO2 retrieved from the ground-based FTIR are compared to XCO2 retrieved from OCO-2, XCH4 and XCO retrieved from the ground-based FTIR are compared to the XCH4 and XCO retrieved from TROPOMI satellite observations. However, the description of the paper lacks scientific significance and originality. Also ,the time series of target gases cover only one year period, so some discussion and conclusions are not representative.

The paper is aim to describe the new ground-based FTIR $XCO_2$, $XCH_4$ and $XCO$ measurements at Xianghe, and to show that the data quality of our FTIR measurement is comply with the TCCON requirement. The site is located in a very polluted area in North China, with no TCCON site at the moment. These measurements are very useful for the climate and air pollution studies as well as related satellite validation.

Specific comments:
1. The aim of the study is to validate satellite data using the FTIR observations, but the paper doesn't describe how to evaluate the accuracy or precision of the FTIR observations.
In the revised version, we reword the title, abstract, introduction and conclusion to make the target of our paper more clear. In fact, the aim of the study is to describe the ground-based FTIR data at Xianghe, and to show that the data quality of our FTIR measurement is comply with the TCCON requirement.

The data quality at Xianghe is assessed by the instrument line shape, the quality of the spectra, the uncertainty of the metadata, the residual of the fitting, and the standard deviation of the retrievals. The setup of the instrument at Xianghe follows the guidance of the TCCON, and in this paper we prove that all these parameters meet the TCCON requirements. In addition, the spectra at Xianghe are analyzed by the standard retrieval code (GGG2014), where the systematic uncertainty (accuracy) of the retrieval is mainly from the spectroscopy. According to the comparison with aircraft or AirCore measurements (Wunch et al., 2015), the systematic correction factors are already implemented in the GGG2014 code to eliminate the systematic uncertainty of $XCO_2$, $XCH_4$ and $XCO$ retrievals (mainly from the spectroscopy uncertainty). The random uncertainties (precision) of the $XCO_2$, $XCH_4$ and $XCO$ retrievals at Xianghe are evaluated by their standard deviations (see Table 1 and Table 2 in the revised version), and their random uncertainties are all within the TCCON reported errors (Wunch et al., 2015).

2. The discussion about day to day variations of Xgas in section 3.3 only use 6-day data, for example in Fig. 10 and 11, so the conclusions about the day to day variation trend of Xgas and the emission source are not reliable and representative.
Thanks for the suggestions. More analyses are added in the revised to make the statement reliable and representative:

Figure 1 (a, b) shows the correlations between the XCO and XCO₂ daily means and between the XCO and XCH₄ daily means at Xianghe. XCO₂ is high in winter and low in summer, and XCH₄ is high in summer and autumn and low in winter. In order to reduce the impact from the seasonal variation, a linear regression model is used to fit the time series of the measurements

$$Y(t) = A_0 + \sum_{k=1}^{3}(A_{2k-1}\cos(2k\pi t) + A_{2k}\sin(2k\pi t)) + \Delta Y(t),$$

where $Y(t)$ is the measurements of XCO₂, XCH₄ or XCO; $A_0$ the mean of the measurements (background), and $A_1$-$A_6$ are the amplitudes of the periodic variations during the year (seasonal variation); $\Delta Y(t)$ is the measurement without background and seasonal variations, representing the day-to-day variation. Note that, we assume there are no trends of these species due to a relatively short time coverage of about one year. Figure 1 (c, d) show the correlations between the ΔXCO and ΔXCO₂ daily means and between the ΔXCO and ΔXCH₄ daily means. The correlation coefficient (R) between XCO and XCO₂ increase from 0.50 to 0.66, and the R between XCO and XCH₄ increase from 0.67 to 0.82. The seasonal variation of ΔXCO₂ still can be observed, but the amplitude is much reduced. There is almost no seasonal variation of ΔXCH₄. Figure 2 shows the correlations in all seasons. It is found that a good correlation between ΔXCO and ΔXCH₄ is found for the whole year, with R values in the range of 0.72-0.87. There is a good correlation (R>=0.85) between ΔXCO and ΔXCO₂ in autumn and winter, and a slightly weak correlation (R=0.47) in spring and (R=0.57) in summer.

[Figure]

Figure 1. The correlation plots between the XCO and XCO₂ and XCH₄ daily means (a, b) and the correlation plots between the ΔXCO and ΔXCO₂, and ΔXCH₄ daily means (c, d) from

FTIR TCCON-type measurements at Xianghe. The dash red line is the linear fit. The N is the number of the measurement days, and R is the correlation coefficient. The error bar is the standard deviation of the measurements in each day. The data are colored with the measurement months.

[Figure]

Figure 2. Upper panels: the correlation plots between the ΔXCO and ΔXCO₂ in four seasons (spring: March, April, May (MAM); summer: June, July, August (JJA); autumn: September, October, November (SON); winter: December, January, February (DJF)). Lower panels: the correlation plots between the ΔXCO and ΔXCH₄ in four seasons. The dash line is the linear fit. The N is the number of the measurement days, and R is the correlation coefficient. The error bar is the standard deviation of the measurements in each day.

In revised version, we add the Lagrangian particle dispersion model version 9.02 (FLEXPART) 10-days backward trajectories for all polluted and clean days. It is assumed that the random distribution of the ΔXCO is symmetric, and the lowest ΔXCO is -36 ppb. Therefore, each day with a ΔXCO>36 ppb is classified as a polluted day, vice versa. In total, we have 28 polluted days and 187 clean days. FTIR measurements show the ΔXCO, ΔXCO₂ and ΔXCH₄ are much larger in the polluted days than those in the clean days (see Table 1).

Table 1. The mean and standard deviation of ΔXCO, ΔXCO₂ and ΔXCH₄ at polluted and clean days.

|  | Polluted days | Clean days |
|---|---|---|
| ΔXCO [ppb] | -9.51±21.10 | 58.40±19.58 |
| ΔXCO₂ [ppm] | -0.64±2.05 | 2.75±2.01 |
| ΔXCH₄ [ppm] | -0.003±0.016 | 0.029±0.019 |

The FLEXPART is able to simulate a large range of atmospheric transport processes, taking mean flow, deep convection, and turbulence into account. The backward running of FLEXPART provides the release–receptor relationship, which is applied to study the source and transport of the observations from a measurement site. In this study, 20000 air particles are released at Xianghe between 10:00 – 14:00 (local time) for days when FTIR measurements are available in the vertical range of surface – 2 km, and a 4-D response function to emission inventory is calculated. The model was driven by the meteorological data from the European Centre for Medium Range Weather Forecast (ECMWF). The residence time of particles in output grid cells describes the sensitivity of the receptor to the source. Figure 3 shows the mean air sources for polluted and clean days. It is found that the

air is mainly from the south and the local polluted region (North China) for the polluted days, and is mainly from the north and remote clean places (Inner Mongolia, Mongolia and Russia) for the clean days.

[Figure]

Figure 3. The mean emission response sensitivities of the air mass at Xianghe (the cross symbol) for polluted (left) and clean (right) days in the vertical range from surface to 2000 m a.s.l. simulated with a 10 day backward run with FLEXPART v9.02.

3. The FTIR measurements need to be very precise and accurate to be useful for satellite validation or model studies, a proper demonstration over a longer period of time is therefore needed for the site. However, the data cover only one year period.

Although only about one-year (June 2018 – July 2019) FTIR measurements are presented in the manuscript (in fact, the measurements are continually operating now), these measurements are still very useful for satellite validation or model studies as the following reasons:

1) We have 215 days' measurements of 15435 retrievals with good data quality between June 2018 and July 2019. As ground-based TCCON-type FTIR spectra are only recorded under the condition of the solar and clear sky, the amount of the measurement days and spectra at Xianghe are very good. As an example, at Bremen (a TCCON site), there are 235 days' measurements of 9473 retrievals during 4 years (2015-2018).

2) As the good time coverage mentioned above, the seasonal variations of $XCO_2$ and $XCH_4$ are well recognized by our FTIR measurements. The variations are compared to other TCCON sites with a similar latitude. Figure 9 shows that the phase of the seasonal variation of $XCO_2$ at Xianghe is close to other sites, but with a slightly larger amplitude of the seasonal variation. However, the phase of the seasonal variation of $XCH_4$ at Xianghe is very different from other sites, which is very interested to model studies. In addition, There is no clear seasonal variation in XCO at Xianghe, but with a much large XCO value together with a large variation. The XCO behavior is quite different compared to other sites. The time series of the FTIR measurements can be applied to study the sources and sinks of $CO_2$, $CH_4$ and CO in this region. For example, the reaction with OH is the main sink of atmospheric $CH_4$ (Rasmussen, et al., 1981). As the OH concentration is high in summer, there is a minimum in $XCH_4$ at Pasadena, Lamont and Karlsruhe. The maximum $XCH_4$ value in summer at Xianghe indicates that there is a strong $CH_4$ source in summer.

3) For satellite validation, the number of co-located FTIR measurements and OCO-2 $XCO_2$ measurements is 28 days, and the numbers of co-located FTIR measurements and TROPOMI $XCH_4$ and XCO measurements are 96 and 70 days, respectively. The number of co-located data pairs are comparable with other validation studies. Wunch

et al., (2017) used the TCCON measurements to validate the OCO-2 satellite XCO2 data for over 2 years. The number of co-located TCCON and OCO-2 data (sum of glint and nadir) pairs is less than 28 at Eureka, Sodankyla, Bialystok, Bremen, Rikubestu, Manaus. For TROPOMI validation, the FTIR measurements at Xianghe is more useful, as the TROPOMI was launched on 13 October 2017. The near real time is only available after June 2018. The FTIR measurements at Xianghe can provide very useful information in a polluted area of North China. Even for the offline reprocess data between November 2017 and June 2019, the co-located TROPOMI and FTIR data at Xianghe is much more than that at some TCCON sites (Eureka, Bremen, Paris, JPL, Darwin, Reunion). For the details, please refer to Lambert et al., (2019). In addition, Fig 12, 13, 14 show that co-located FTIR and satellite data pairs are distributed evenly in all seasons.

4. In Line 25 Page 11, "The retrieved TROPOMI CO data is in the unit of total column density (molecules/cm2), so we converted them to to XCO (ppb) values for comparison with FTIR XCO measurements", there should be a short description of the method and cite a reference.
OK, added in the revised version.

5. In Line 12 Page 12, "Regular HCl cell measurements show that the ME loss is within 2% and the PE remains within 0.02 rad", the conclusion is not consistent with the ME results in Fig. 2.
Corrected. "Regular HCl cell measurements show that the ME loss is within 5% and the PE remains within 0.02 rad".

6. There are no unit for the mean and std value in the Fig. 12, Fig .13 and Fig. 14.
The units for the mean and std are the same as shown in the y axis. Added in the revised version.

7. In Line 14 Page 1, "The rapid economic growth of China has contributed to 30% of the global total carbon dioxide (CO2) emissions from fossil fuel consumption and cement production (Jackson et al., 2017)", the exact contribution is about 28.5% according to the results in Jackson et al., 2017.
Corrected

*Black: referee's comments* *red: authors' answers*
*First of all, we want to thank the referee for the detailed analysis of our paper.*
*For the details, please look into the paper with keeping track of changes.*

Anonymous Referee #2
General comments The manuscript "A new site: ground-based FTIR XCO2, XCH4 and XCO measurements at Xianghe, China" by Yang et. al. describes a data set of total column dry air mole fractions of carbon dioxide, methane and carbon monoxide derived from near infrared, solar absorption Fourier transform spectroscopy using methods similar to those of the Total Carbon Column Observing Network (TCCON). The described dataset covers a year of measurements and therefore captures a single seasonal cycle. The manuscript follows a logical narrative, but in some areas would benefit from additional copy editing.

Data of this type, from an urban area of China, are likely to be of value to the satellite and model validation communities and I would encourage the publication of this manuscript subject to a number of clarifications and modifications as detailed below.

Specific comments
Reference is made to TCCON throughout the manuscript, however it is not made explicitly clear that the Xianghe site is not currently affiliated to TCCON. This should be stated from the start including what steps are required for the site to become affiliated with TCCON if that is the objective. Additionally, statements such as "this study shows that the Xianghe data comply with the TCCON specifications" at page 12 line 30 are considered quite strong and possibly misleading as, for example, the TCCON requirement for an in-situ validation is not covered in this work. Care should also be taken when making comparison to the TCCON accuracy/precision requirements, throughout the manuscript reference is made to "the 0.8 ppm (SZA less than 80 ∘) retrieval accuracy of TCCON XCO2." (e.g. at P4 L23), this is actually the target for site-to-site biases within the TCCON. The actual single instrument/site precision target for TCCON is 0.1% i.e. 0.4 ppm for XCO2. Precision estimates within the manuscript are based on the standard deviations of daily retrievals of each of the species. This may result in an overly pessimistic estimate as the daily means are potentially influenced by a number of factors such as a diurnal cycle, the influence of local sources in such a heavily urbanized area and a residual airmass dependence of the retrievals as pressure broadening is not accounted for in the GGG2014 spectroscopy. Better and more representative results may be obtained by limiting the windows over which standard deviations are calculated to a smaller range of solar zenith angles or shorter periods.
Thanks for the suggestions.
The aim of the study is to describe the ground-based FTIR data at Xianghe, and to show that the data quality of our FTIR XCO$_2$, XCH$_4$ and XCO measurements is comply with the TCCON requirement (except an in-situ validation). The site has not been affiliated to TCCON yet. Therefore, the main concern of this work is to demonstrate the quality of the spectra at Xianghe and to be ready to join TCCON in near future. Steps are added in the beginning to make the target of this paper clear.

We adapt the description about the precision of the TCCON XCO$_2$ in the revised paper. According to Pollard et al., (2017), the precision target for TCCON is 0.1% (~0.4 ppm) for XCO$_2$ to meet the model requirement (Olsen and Randerson, 2004). However, the precision of the TCCON XCO$_2$ is estimated to be 0.2% (~0.8 ppm) based on the perturbation of the GGG2014 inputs (Wunch et al., 2015).

To estimate the precision of the retrievals at Xianghe, we limit the windows over which standard deviations are calculated to a smaller range of solar zenith angle (SZA less than 30°) to reduce the potential influences, such as a diurnal cycle, the influence of local sources in such a heavily urbanized area and a residual airmass dependence of the retrievals as pressure broadening is not accounted for in the GGG2014 spectroscopy. We select all the days when at least 5 measurements are available. Table 1 is updated as follows

Table 1. The mean standard deviation (STD) of $XCO_2$, $XCH_4$ and XCO of the remaining data after solar intensity (SI) filtering for each day and the percentage of the remaining spectra amount from 14 June 2018 to 31 December 2018. Only measurements with SZA less than 30°are selected.

| β (%) | γ (%) | STD($XCO_2$) (ppm) | STD($XCH_4$) (ppm) | STD(XCO) (ppm) | Percentage of remaining spectra (%) |
|---|---|---|---|---|---|
| 85 | 10 | 0.749 | 0.0042 | 3.032 | 89.5 |
| 85 | 5 | 0.723 | 0.0036 | 2.954 | 88.9 |
| 85 | 0 | 0.667 | 0.0035 | 2.997 | 88.1 |
| 90 | 10 | 0.651 | 0.0036 | 2.904 | 86.6 |
| 90 | 5 | 0.648 | 0.0035 | 2.908 | 85.8 |
| 90 | 0 | 0.496 | 0.0035 | 2.559 | 84.7 |
| 95 | 10 | 0.450 | 0.0029 | 2.421 | 78.5 |
| 95 | 5 | 0.447 | 0.0029 | 2.436 | 77.1 |
| 95 | 0 | 0.423 | 0.0026 | 2.438 | 75.2 |

In order to reduce the variation of the measurements and to keep as many as useful measurements, we choose the β=90%,γ=0% as the criteria for the solar intensity filtering. The mean STD of $XCO_2$, $XCH_4$ and XCO are 0.496 ppm, 0.0035 ppm and 2.559 ppb, respectively, and there are 84.7% spectra remained. Note that the mean STD of $XCO_2$ is 0.12%, which is slightly worse than the target of the TCCON $XCO_2$ precision of 0.1%, but it is better than the estimated uncertainty of 0.2%. To evaluate the precision of the retrievals at Xianghe, we compare the STD of $XCO_2$ measurements at Xianghe with several standard TCCON sites with similar latitude (Lamont, Karlsruhe, Pasadena, Rikubetsu, Tsukuba, Saga, Orleans and Anmeyondo) in Table 2. The STD of $XCO_2$ at Xianghe is 0.496 ppm, which is less than most sites except Karlsruhe and Orleans. The precision of $XCO_2$ measured at Xianghe is comparable with other TCCON sites.

Table 2. The mean standard deviation (STD) of $XCO_2$ for the measurements at each day with the solar zenith angle less than 30°, together with total numbers of the days and the STD less than 0.1% for $XCO_2$ since January 1, 2017 at Anmeyondo, Karlsruhe, Lamont, Orleans, Pasadena, Rikubetsu, Saga, Tsukuba. At Xianghe, the measurements are between 14 June 2018 to 31 December 2018 after the SI filtering with the β=90%,γ=0%.

| Sites | Latitude (N) | N (total) / N (STD < 0.1%) | Percentage (STD< 0.1%) | STD($XCO_2$) (ppm) |
|---|---|---|---|---|
| Anmeyondo | 36.5 | 9 / 0 | 0 | 0.925 |
| Karlsruhe | 49.1 | 73 / 57 | 78 | 0.311 |
| Lamont | 36.6 | 287 / 0 | 0 | 0.718 |
| Orleans | 48.0 | 49 / 41 | 94 | 0.375 |
| Pasadena | 34.1 | 280 / 38 | 14 | 0.524 |
| Rikubetsu | 40.0 | 33 / 8 | 24 | 0.507 |

| Saga | 33.2 | 192 / 34 | 18 | 0.589 |
| Tsukuba | 43.5 | 106 / 31 | 29 | 0.506 |
| Xianghe | 39.7 | 19 / 7 | 37 | 0.496 |

In addition, the table 3 is updated based on the measurements with the SZA less than 30°. For the AC+DC mode, the STD of $XCO_2$ with only SNR filtering is the best. If we apply both SNR filtering and SI filtering, many reasonable retrievals will be filtered out, which have bad SI flags but have been corrected by the DC correction procedure. Therefore, for the period with AC+DC mode, only the SNR filtering is applied.

Table3. Average $XCO_2$ daily standard deviation after (a) no filtering, (b) SNR filtering, (c) both SNR filtering and SI filtering during the AC mode and AC+DC mode periods. The unit is ppm. Only those measurements with SZA less than 30° are considered.

|  | AC mode | AC + DC mode |
| --- | --- | --- |
| a | 1.19 | 1.06 |
| b | 0.94 | 0.53 |
| c | 0.57 | 0.88 |

In sub-section 2.2 it would also be useful to present some example plots of ME and PE as a function of OPD, and as PE can have a maximum value at an OPD other than the maximum, consider presenting a time series of maximum PE. Also, LINEFIT 14.5 can be used in a number of modes, it should be explicit which mode is used.
Thanks for the suggestions.
We plot the ME and PE as a function of OPD, together with the maximum ME loss and maximum PE deviation at Xianghe in the revised version, which is showed in Figure 1. The ME has a maximum loss at the MOPD while PE has a maximum deviation at about 20 cm (positive value) or the MOPD (negative value).

These ILS parameters are retrieved by LINEFIT 14.5 using 13 HCl microwindows under non-vacuum status. The degree of freedoms for signal (DOFS) of anodization and phase are about 4.1 and 4.2, respectively.

[Figure]

Figure 1. The modulation efficiency (ME, left panel) and phase error (PE, right panel) along the optical path difference (OPD) at Xianghe. The purple dots are the maximum ME loss (left) and the maximum PE deviation (right).
The sub-section on signal-to-noise ratio would benefit from a discussion about how SNR is calculated as there are a number of possible methods. Making this explicit would make it easier to make comparisons between sites.
Thanks for the suggestion. More information has been added in the revised paper.
We calculate SNR as the ratio between the maximum intensity of the spectrum in the spectral range of 3800-11000 cm$^{-1}$ and the noise level. The standard deviation of the intensity

between 2350 and 2450 cm$^{-1}$ is calculated as the noise level, since no signal is recorded in this window.

$$SNR = \max(I) / STD(noise).$$

In the paragraph describing satellite missions that can be validated by this type of measurement in the introduction, GOSAT is notable by its absence, and as one of the first satellites dedicated to measuring greenhouse gases probably deserves to be mentioned. Added.

Ongoing measurements of this type are likely to be of interest to a number of users and so a statement about whether future measurements will be added to the referenced dataset, or the TCCON archive if the site should become affiliated to the network, would be useful. Thanks for the suggestion.
The data used in this study can be accessed by the data DOI, and the public can download them for their interests, such as satellite validation and model inversions. In the near future, we hope to join the TCCON successfully via the institutional agreement and upload the ongoing measurements to the TCCON archive.

Technical corrections
P2 L4-8 Whilst not strictly necessary, it would be useful to provide more context by comparing the current values to their pre-industrial levels and/or stating the current rate of increase.
Done

P2 L10 Provide the radiative forcing of CO2 as a comparison.
Done

P2 L13-14 Try to clarify the distinction that is being made here, i.e. are the two main types of measurement in-situ and remote sensing, or ground-based and satellite?
To avoid the confusion, we remove this sentence in the revised version.

P3 L1-2 Inclusion of the sentence "To increase the precision of the retrievals, the spectra are cloud filtered based on the separate direct solar irradiation measurements." fragments the description of the manuscript outline, considering removing it.
Done.

P3 L33 After "NIR InGaAs" insert spectra or measurements.
Done

P3 L26 - P4 L7 These two paragraphs highlight the importance of consistent representation of scalars and their vectors e.g. m and meters are both used and sometimes there is a space between the scalar and vector and sometimes not.
Corrected

P5 Eq.1 Appears to have been inserted at the wrong point in the manuscript.
Corrected

P5 L16 Please give details of these sensors, particularly the type and precision/accuracy of the pressure sensor which is very important for the accuracy of trace gas retrievals.

Added

P5 L20 Does the first "a priori partial column" refer to the tropospheric component? This should be made more explicit.
Done.

P6 L10 Consider using affected instead of infected.
Done.

P9 Eq.4 The vector I is not defined.
Added

P11 L8, L14 and L31 The Landgraf et al references do not contain a year.
Corrected

Table 2. AC+DC mode column, it would be expected that additional filtering would result in a lower value for SNR+SI (row c) than for SNR (row b) on its own. This doesn't appear to be the case.
For the AC+DC mode, the STD of $XCO_2$ with only SNR filtering is the best. If we apply both SNR filtering and SI filtering, many reasonable retrievals will be filtered out, which have bad SI flags but have been corrected by the DC correction procedure. Therefore, for the period with AC+DC mode, only the SNR filtering is applied.

Reference:
Olsen, S. C. and Randerson, J. T.: Differences between sur-face and column atmospheric CO2and implications for car-bon cycle research, J. Geophys. Res.-Atmos., 109, D02301,https://doi.org/10.1029/2003JD003968, 2004.

[revised manuscript text omitted]

---

## Referee Report (RR1)

**Review of "New ground-based Fourirer-transform near-infrared solar absorption measurements of XCO2, XCH4 and XCO at Xianghe, China" by Yang et.al**

**General comments** The manuscript "New ground-based Fourirer-transform near-infrared solar absorption measurements of XCO2, XCH4 and XCO at Xianghe, China" by Yang et.al is a revised version of the manuscript "A new site: ground-based FTIR XCO2, XCH4 and XCO measurements at Xianghe, China" following the discussion stage of the peer review process and describes a data set of total column dry air mole fractions of carbon dioxide, methane and carbon monoxide derived from near infrared, solar absorption Fourier transform spectroscopy using methods similar to those of the Total Carbon Column Observing Network (TCCON). The described dataset covers a year of measurements and therefore captures a single seasonal cycle.

The revised version of the manuscript has taken account of the reports of both reviewers and taken steps to address them. Subject to a few technical changes below, I would recommend publication in ESSD.

**Specific comments**

**Response to reviewer 1** The authors have revised the manuscript in order to make the aim of the work clearer. They have also expanded the sub-section discussing the results of their retrievals to inlcude XCO tracer based analysis and time-series decomposition in order to identify potential sources of relatively polluted and clean airmasses as identified by their observations. This additional material addresses the reviewer's concern that the 'the description of the paper lacks scientific significance and originality' and is likely sufficient to satisfy the aims and scope for a data description paper.

**Response to reviewer 2** The authors have made changes to the manuscript to clarify the relationship between the Xianghe site and the Total Carbon Column Observing Network (TCCON). It will be good to see the time-series extended and incorporated into TCCON in the future. Other specific comments have been addressed satisfactorily.

**Technical corrections**

P8 in the discussion of seasonal variations include both the month and year when describing maximum and minimum values to make it clear that these values relate to a specific cycle rather than every cycle.

P9 L21 the choice for the vertical range should be explained.

P14 L9 "appear very" is a qualitative statement consider changing to e.g. "are demonstrated to be".